# Baicalein Activates Parkin-Dependent Mitophagy through NDP52 and OPTN

**DOI:** 10.3390/cells11071132

**Published:** 2022-03-28

**Authors:** Po-Yuan Ke, Chih-Wei Chang, Yuan-Chao Hsiao

**Affiliations:** 1Department of Biochemistry & Molecular Biology and Graduate Institute of Biomedical Sciences, College of Medicine, Chang Gung University, Taoyuan 33302, Taiwan; will0521@gmail.com (C.-W.C.); c11kkh@gmail.com (Y.-C.H.); 2Liver Research Center, Chang Gung Memorial Hospital, Taoyuan 33305, Taiwan

**Keywords:** autophagy, mitophagy, baicalein, selective autophagy, parkin, cargo receptor

## Abstract

The elimination of intracellular components by autophagy maintains metabolic homeostasis and is a quality-control pathway that enables organelle regeneration. Mitophagy is a type of selective autophagy that regulates mitochondrial turnover, and the dysregulation of mitophagy has been implicated in the pathogenesis of liver diseases. However, the detailed molecular mechanism underlying mitophagy regulation in liver cells remains unclear, and the small molecules that may potentially modulate hepatic mitophagy are still unavailable. Here, we report that baicalein, a flavonoid extracted from *Scutellaria baicalensis*, induces the entire autophagy that proceeds through the autolysosome maturation stage in human hepatoma cells. In addition, baicalein-induced autophagy is demonstrated to target mitochondria for degradation. Further studies show that baicalein triggers the translocation of Parkin and TBK1 to mitochondria to induce mitophagy. Moreover, the phosphorylation of TBK1 at Ser172 and ubiquitin at Ser65 is shown to trigger mitophagy in baicalein-treated cells. Furthermore, two specific autophagy cargo receptors, NDP52 and OPTN, that function in baicalein-activated mitophagy are identified. Taken together, these findings not only delineate the molecular process of Parkin-dependent mitophagy in liver cells, but also reveal baicalein as a novel inducer of hepatic mitophagy.

## 1. Introduction

Autophagy is an evolutionarily conserved process that catabolizes intracellular components via lysosomal degradation [1,2]. Targeting intracellular components for autophagic degradation not only eliminates damaged organelles but also enables nutrient recycling for metabolic homeostasis and organelle regeneration. Defective autophagy has been convincingly shown to be associated with the development of several human diseases, including neurodegenerative diseases, cancer, infectious diseases, and liver diseases [3,4]. Autophagy is a stepwise process of vacuole biogenesis that is initiated with the nucleation of the isolation membrane (IM)/phagophore, its elongation and transition from a cup shape into a double-membrane autophagosome, and finally, the fusion of this autophagosome with a lysosome to form a mature autolysosome in which lysosomal proteases degrade IM/phagophore-sequestrated components [5,6]. The successful completion of autophagy relies on the coordinated action of several autophagy-related genes (ATGs), signaling cascades, nutrient-sensing molecules, and vesicle-trafficking pathways [7,8,9]. In cells deprived of nutrients, the repression of mammalian target of rapamycin (mTOR), a pivotal serine/threonine kinase hub for regulating cell metabolism, induces the translocation of the unc-51-like kinase (ULK) complex (ULK1/2, ATG101, ATG13, and RB1-inducible coiled-coil 1 (RBCC1)) from the cytosol to an endoplasmic reticulum (ER) membrane-reconstituted compartment. This translocation triggers the recruitment and activation of the class III phosphatidylinositol-3-OH kinase (PI(3)K) complex (Vps34/PI3KC3, Beclin 1, ATG14, and Vps15), thus promoting phosphatidylinositol-3-phosphate (PtdIns(3)P) synthesis. The generated PtdIns(3)P then recruits double-FYVE-containing protein 1 (DFCP1) and WD-repeat domain PtdIns(3)P-interacting (WIPI) family proteins to reconstitute the IM/phagophore. Subsequently, ubiquitin-like (UBL) conjugation systems induce the expansion and elongation of the IM/phagophore to eventually form a closed double-membranous autophagosome. A signaling cascade consisting of ATG7 (E1) and ATG10 (E2) induces the covalent-linking of ATG12 to ATG5, forming the ATG12-ATG5 conjugate that in turn binds to ATG16L, leading to an ATG12-ATG5-ATG16L trimeric complex formation. To target ATG8/LC3 family proteins to the autophagosomal membrane, the C-termini of ATG8/LC3 family proteins are first cleaved by ATG4 family proteins to form ATG8/LC3-I, which is conjugated to phosphatidylethanolamine (PE) by ATG7 (E1) and ATG3 (E2), forming PE-conjugated ATG8/LC3 (referred to as ATG8/LC3-II). Together, the ATG12-ATG12-ATG16 complex and ATG8/LC3-II promote the expansion of the IM/phagophore and the maturation of the autophagosomes. Finally, the autophagosomes fuse with lysosomes to form mature autolysosomes with acidic proteases capable of degrading engulfed material.

In addition to bulk and nonselective degradation, autophagy selectively eliminates aggregated proteins and damaged organelles through selective autophagy [10,11,12,13]. Several types of cargo receptors are activated for selective autophagy; these proteins include p62/sequestosome 1 (p62/SQSTM1), the neighbor of BRCA1 (NBR1), nuclear domain 10 protein 52 (NDP52, also known as CALCOCO2), optineurin (OPTN), Tax1-binding protein 1 (TAX1BP1), and BCL2/adenovirus E1B 19 kDa protein-interacting protein 3-like (BNIP3L, also known as Nix), most of which harbor a ubiquitin-associated domain (UBA) for binding to polyubiquitinated cargoes [13]. Furthermore, ATG8/LC3-interacting regions (LIRs) in these cargo receptors mediate the interaction between the cargo receptors and ATG8/LC3 located on the autophagosomal membrane, thus targeting the cargoes to autophagosomes [14]. Selective autophagy is a quality-control mechanism through which defective organelles are removed in a process called “organellophagy” to maintain the integrity of organelles and promote the regeneration of organelles [12,15].

Mitophagy is a form of organellophagy that specifically degrades deformed mitochondria to promote mitochondrial turnover [16,17,18]. Depolarization of the mitochondrial membrane leads to defects in the import of molecules into mitochondria and interferes with the proper proteolysis of PTEN-induced putative kinase 1 (PINK1), causing constitutive PINK1 stabilization [19]. Stabilized PINK1 is translocated to the mitochondrial outer membrane (MOM), where it recruits Parkin and phosphorylates ubiquitin at serine residue (Ser) 65 [20,21,22]. Activated Parkin subsequently promotes the ubiquitination of MOM-bound proteins through the enzymatic activity of its specific E3 ubiquitin ligase, recruiting cargo receptors, such as OPTN and NDP52, for mitochondrial elimination through PINK1/Parkin-dependent mitophagy [17,23]. Moreover, TANK-binding kinase 1 (TBK1) specifically engages PINK1/Parkin-mediated mitophagy by phosphorylating p62/SQSTM1 and OPTN at serine residues [24,25,26]. Despite PINK1/Parkin-dependent mitophagy, several mitochondrial proteins, such as BCL2/adenovirus E1B 19 kDa protein-interacting protein 3 (BNIP3), BNIP3L/Nix, FUN14 domain-containing 1 (FUNDC1), and prohibitin 2 (PHB2), serve as cargo receptors in mitophagy [27,28]. Notably, the molecular mechanism of PINK1/Parkin-mediated mitophagy was initially identified in ectopically Parkin-overexpressing HeLa cells (lacking endogenous Parkin) treated with mitochondrial oxidative phosphorylation uncoupling agents, such as carbonyl cyanide m-chlorophenyl hydrazone (CCCP). In fact, little is known about mitophagy regulation under physiological conditions, and whether PINK1/Parkin-dependent mitophagy solely regulates mitochondrial turnover in other cell types and tissues, such as the liver, remains to be determined through further investigation. In the past decade, the precise regulation of mitophagy has been shown to maintain liver function, and dysregulated mitophagy has been reported to participate in the pathogeneses of liver diseases [29,30,31], suggesting that the proper modulation of mitophagy may be exploited to treat liver diseases [32]. However, the small compounds needed to manipulate hepatic mitophagy are still rare and urgently needed.

Baicalein (5,6,7-trihydroxyflavone) is a flavonoid extracted from the Chinese herb *Scutellaria baicalensis* [33,34]. Baicalein has been shown to have anti-inflammatory [35], antioxidant [36], hepatoprotective [37,38], neuroprotective [39], anticancer [40], and antiviral [41] effects. In addition, baicalein has been demonstrated to suppress cell proliferation [42,43], induce cell apoptosis [44,45], and trigger cell cycle arrest [46] in different cancer cell models, suggesting that baicalein can potentially be used to treat cancer. Specifically in liver cancer, baicalein has been shown to induce protective autophagy by increasing ER stress in hepatocellular carcinoma (HCC) cells, and the suppression of autophagy may increase the apoptosis rate of HCC cells [44]. In contrast to its role in promoting autophagy, baicalein has been shown to suppress autophagy by targeting guanosine triphosphatase (GTPase) and thus enhance HCC cell chemosensitivity [47]. Moreover, baicalein has been reported to protect the liver from carbon tetrachloride (CCl_4_)- and acetaminophen (APAP)-induced liver injury by altering autophagy [38,48]. Collectively, these studies have implicated baicalein in differentially exerted hepatotoxicity and anti-HCC processes through its modulation of hepatic autophagy. However, the details on the molecular mechanism underlying baicalein regulation of hepatic autophagy, particularly in HCC cells, remain unclear, and further investigation is needed.

In this study, we found that baicalein activated complete autophagy through the autolysosome maturation stage and enhanced autophagic flux. In particular, our study showed that baicalein-induced autophagy engulfed mitochondria and promoted mitochondrial turnover. Using the mitophagy reporters Mito-QC and MT-Keima, we found that baicalein induced mitolysosome formation and enhanced mitophagic flux. We further demonstrated that baicalein induced the mitochondrial translocation of Parkin, the stabilization of PINK1 in mitochondria, and the phosphorylation of ubiquitin at Ser65. Moreover, baicalein promoted the phosphorylation of TBK1 at Ser172 and triggered the translocation of TBK1 to the mitophagic process. Furthermore, the specific recruitment of NDP52 and OPTN to mitochondria in baicalein-treated cells suggested that these two cargo receptors were involved in baicalein-induced Parkin-dependent mitophagy. Finally, by interfering with autophagy and NDP52- and OPTN-dependent mitophagy through gene knockout, baicalein-induced mitochondrial turnover was dramatically suppressed. Taken together, our study results comprehensively illustrated the molecular process of baicalein-induced autophagy and mitophagy. We specifically revealed that baicalein activated Parkin-dependent mitophagy through NDP52 and OPTN to promote mitochondrial turnover. In conclusion, our findings not only decipher the detailed molecular mechanism underlying how baicalein activates autophagy in liver cells, but also imply that baicalein is a promising inducer of hepatic mitophagy.

## 2. Materials and Methods

### 2.1. Cell Culture, Reagents, and Antibodies

The Huh7 and Huh7.5 human HCC cell lines were kindly provided by Francis Chisari (The Scripps Research Institute, San Diego, CA, USA) and Charles M. Rice (Rockefeller University, New York, NY, USA). HepG2 (HB-8065), an HCC cell line, and 293T (CRL-3216), a cell line derived from human embryonic kidney 293 cells transformed with SV40T antigen, were purchased from the American Type Cell Collection (ATCC, Manassas, VA, USA). The 293T, HepG2, Huh7, and Huh7.5 cells were cultured in DMEM (Thermo Fisher Scientific, Waltham, MA, USA) containing 10% fetal bovine serum (FBS) (Thermo Fisher Scientific) and 1% nonessential amino acids (NEAAs) (Thermo Fisher Scientific) at 37 °C in a 5% CO_2_ atmosphere. Baicalein (465119) was purchased from Sigma–Aldrich (St. Louis, MI, USA), and bafilomycin A1 (BAF-A1, BML-CM110-0100), puromycin (BML-GR312-0050), and CCCP (BML-CM124-0500) were purchased from Enzo Life Sciences (Farmingdale, NY, USA). Rabbit anti-LC3B (L7543), rabbit anti-SQSTM1 (P0067), and mouse anti-β-actin (A5441) antibodies were purchased from Sigma–Aldrich. Rabbit anti-TOM20 (SC-11415), mouse anti-TOM20 (SC-17764), mouse anti-COX IV (SC-376731), and rabbit anti-HSP60 (SC-13966) were purchased from Santa Cruz Biotechnology (Dallas, TX, USA). Rabbit anti-ATG3 (3415), anti-ATG5 (12294), anti-ATG7 (8558), anti-ATG16 (8089), anti-TBK1 (3504), anti-TAX1BP1 (5105), anti-BNIP3L (12396), anti-Mitofusin-1 (MFN1) (14739), anti-Mitofusin-2 (MFN2) (9482), and anti-phospho-TBK1 (Ser172) (5483) were purchased from Cell Signaling Technology (Danvers, MA, USA). Rabbit anti-NDP52 (GTX115378) and rabbit anti-OPTN (A301-831A) were purchased from GeneTex (Irvine, CA, USA) and Bethyl Laboratories (Montgomery, TX, USA), respectively. Mouse anti-mitochondrially encoded cytochrome C oxidase II (MTCO2) (ab110258), anti-Parkin (ab15494), anti-PINK1 (ab23707), and rabbit anti-phospho-ubiquitin (Ser65) (AB S1513-I) were purchased from Abcam (Cambridge, UK) and Merck Millipore (Burlington, MA, USA). Goat anti-rabbit Alexa Fluor 488 (A-11008), anti-mouse Alexa Fluor 488 (A-11001), anti-rabbit Alexa Fluor 555 (A-21428), anti-mouse Alexa Fluor 555 (A-21422), anti-rabbit Alexa Fluor 647 (A-21244), anti-mouse Alexa Fluor 647 (A-21235) secondary antibodies, Hoechst 33342 (H1399), and 4′,6-diamidino-2-phenylindole (DAPI, D21490) used for IF were purchased from Thermo Fisher Scientific. Goat anti-rabbit and anti-mouse horseradish peroxidase (HRP)-conjugated secondary antibodies (A0545 and A9044, respectively) used for Western blotting were purchased from Sigma–Aldrich. Paraformaldehyde (PFA), 8% in water, EM grade (157-8); glutaraldehyde (GA), 25% in water, EM grade (16220); 0.4 M sodium cacodylate buffer (CAB) (11655); osmium tetroxide (OsO4), 4% in water (19190); uranyl acetate (22400); lead citrate (17800); and a low-viscosity embedding medium Spurr’s resin kit (14300) used in sample preparation for electron microscopy were purchased from Electron Microscopy Sciences (Hatfield, PA, USA). The K_4_[Fe(CN)_6_] (P8131) and ethanol (1.00983.2500) used in sample preparation for electron microscopy (EM) were purchased from Sigma–Aldrich and Merck Millipore, respectively. The poly-D-lysine (A3890401) used for correlative light and electron microscopy was purchased from Thermo Fisher Scientific. The nutrient starvation medium containing 20 mM Hepes, pH 7.4, 140 mM NaCl, 1 mM CaCl_2_, 1 mM MgCl_2_, 5 mM glucose, and 1% bovine serum albumin (BSA) was prepared as previously described [49].

### 2.2. Construction of Expression Plasmids

The gene fragments harboring GFP-LC3, RFP-LC3, and RFP-GFP-LC3 were cleaved from pEGFP-LC3 (#21073, Addgene (Watertown, MA, USA)), pmRFP-LC3 (#21075, Addgene), and ptfLC3 (#21074, Addgene) plasmids and subcloned into pTRIP-GFP lentiviral expression plasmids (kindly provided by Charles M. Rice), generating pTRIP-GFP-LC3, pTRIP-RFP-LC3, and pTRIP-RFP-GFP-LC3, respectively. To construct pMito-GFP, pMito-RFP, and pMito-miRFP670 expression plasmids, GFP (from pEGFP-C1), RFP (from pmRFP-LC3), and miRFP670 (from pmiRFP670-N1, #79987, Addgene) gene fragments were subcloned into Mito-PAGFP (#23348, Addgene) to replace PAGFP. Then, the gene fragments containing Mito-GFP, Mito-RFP, and Mito-miRFP670i were subcloned into pTRIP-GFP plasmids to generate pTRIP-Mito-GFP, pTRIP-Mito-RFP, and pTRIP-Mito-miRFP670. A gene fragment of human fission 1 (amino acids 101–152) was synthesized (MDBio, Taiwan) and subcloned into ptfLC3 to replace LC3, producing the pMito-QC (RFP-GFP-FIS1 (101–152)) expression plasmid. The polymerase chain reaction (PCR) amplified the MT-Keima gene fragment from the mitophagy Keima-red-Parkin plasmid (AM-V0259M, MBL, Japan), and it was subcloned into pEGFP-C1 (TaKaRa, Japan) to generate a pMT-Keima plasmid. The gene fragments of Mito-QC and MT-Keima were then subcloned into pTRIP-GFP, generating pTRIP-Mito-QC and pTRIP-MT-Keima, respectively. To construct pRFP-Parkin, p3XFLAG-Parkin, and pmTagBFP2-PArkin, a YFP gene fragment in YFP-Parkin (#23955, Addgene) was excised and replaced with RFP, 3XFLAG tags, and mTagBFP2 (from pmTagBFP2-C1, #79987, Addgene). PCR-amplified human SQSTM1, NBR1, NDP52, OPTN, TAX1BP1, BNIP3L, and TBK1 were subcloned into pEGFP-C1 to construct pEGFP-C1-SQSTM1, pEGFP-C1-NBR1, pEGFP-C1-NDP52, pEGFP-C1-OPTN, pEGFP-C1-TAX1BP1, pEGFP-C1-BNIP3L, and pEGFP-C1-TBK1, respectively. YFP-Parkin, RFP-Parkin, 3XFLAG-Parkin, mTagBFP2-Parkin, GFP-SQSTM1, GFP-NBR1, GFP-NDP52, GFP-OPTN, GFP-TAX1BP1, GFP-BNIP3L, and GFP-TBK1 were amplified by PCR and subcloned into pLenti-EF-Blast (a plasmid engineered with lentiCas9-Blast (#52962, Addgene) lacking the Cas9 gene) to generate pLenti-EF-YFP-Parkin, pLenti-EF-RFP-Parkin, pLenti-EF-3XFLAG-Parkin, pLenti-EF-mTagBFP2-Parkin, pLenti-EF-GFP-SQSTM1, pLenti-EF-GFP-NBR1, pLenti-EF-GFP-NDP52, pLenti-EF-GFP-OPTN, pLenti-EF-GFP-TAX1BP1, pLenti-EF-GFP-BNIP3L, and pLenti-EF-GFP-TBK1. The sequences of PCR-amplified fragments for gene cloning in this study were confirmed by the automatic DNA sequencing method. The detailed information regarding plasmid construction for this study is available from the corresponding author upon reasonable request.

### 2.3. Establishment of Stable Reporter Cells by Lentiviral Gene Delivery

Lentiviruses were generated following a previously described procedure [50]. For the production of pTRIP-derived lentiviruses, the pTRIP-cDNA transfer plasmid; pCMVRd8.91 (provided by the RNAi core facility, Academia Sinica, Taipei, Taiwan), a packaging plasmid containing the gag, pol, and rev genes; and pMD.G (provided by the RNAi core facility, Academia Sinica), a VSV-G envelope-expressing plasmid, were cotransfected at a ratio of 1:1:1 into 293T cells. Seventy-two hours post-transfection, the culture supernatants containing infectious viruses were harvested and filtered through a 0.45-μm filter and immediately used for transduction. The packaging of pLenti-EF-related lentiviruses was as described above, except that the packaging plasmids, psPAX2 (#12260, Addgene) and pMD2.G (#12259, Addgene) were used for cotransfection. For the establishment of stable reporter cells by lentiviral gene delivery, Huh7 cells were seeded in a 12-well plate at a density of 1 × 10^4^ cells/well 18 h before transduction. The cells were then incubated with lentiviruses in the presence of 8 μg/mL polybrene and spin-inoculated at 1100× *g* for 4 h at 25 °C. Twenty-four hours after transduction, the lentiviruses were removed and replenished with fresh DMEM supplemented with 10% FBS and 1% NEAAs.

### 2.4. SDS-PAGE, Western Blotting, and Purification of Mitochondria

For the analysis of protein expression, cells were harvested with RIPA buffer (50 mM Tris, pH 7.4; 150 mM NaCl; 1% NP-40; 0.5% DOC; and 0.1% SDS) supplemented with a protease inhibitor cocktail (Roche, Switzerland) and a phosphatase inhibitor cocktail (Sigma–Aldrich). The cell lysate was then separated by centrifugation at 12,000× *g* at 4 °C for 10 min, and the supernatant containing the soluble protein fraction was collected. The concentration of the protein was determined with a Bradford protein assay (Bio-Rad Laboratories, Hercules, CA, USA). Equal amounts of protein were separated by SDS-PAGE at the appropriate percentage and then electrophoretically transferred onto a 0.45-μm PVDF membrane (Millipore). The transfer membrane was subsequently blocked by incubation with 3% nonfat milk in TBST buffer (20 mM Tris, pH 7.4; 150 mM NaCl; and 0.1% Tween-20) for 30 min and then incubated with specific primary antibodies. After extensive washing, the membrane was incubated with the cognate HRP-conjugated secondary antibodies. The signals of the proteins on the membrane were detected with an enhanced chemiluminescence (ECL) kit (Millipore) as described in the manufacturer’s instruction manual. Isolation of mitochondrial fraction was performed with the Mitochondria Isolation Kit for Cultured Cells (89874, Thermo Fisher Scientific) as described in the manufacturer’s instruction manual. The purified fraction of mitochondria was solubilized with RIPA buffer containing protease and phosphatase inhibitors and then immediately used for SDS-PAGE and Western blotting.

### 2.5. Immunofluorescence (IF), Confocal Microscopy, and Fluorescence-Activated Cell Sorting (FACS) Analysis

For IF and confocal microscopy of fluorescence-tagged autophagy and mitophagy reporter cells, the cells were washed twice with phosphate-buffered saline (PBS), fixed with 4% PFA in PBS for 30 min at room temperature (RT), and stained with Hoechst 33342 at RT for 15 min. After three washes with PBS, the cells were observed with confocal microscopy. For the analysis of endogenous lysosome- and mitochondria-associated proteins through immunostaining and confocal microscopy, the cells were washed twice with PBS and fixed with 4% PFA in PBS for 30 min at RT, followed by three washes with PBS. Then, the cells were blocked with 2% goat serum in PBS for 1 h at RT and incubated with specific primary antibodies at 4 °C for 12 h, followed by incubation with the cognate Alexa Fluor-conjugated secondary antibodies and DAPI at RT for 1 h. The images were captured with a Plan-Apochromat 63×/1.4 Oil DIC M27 objective of an LSM780 laser scanning confocal microscope and analyzed with Zen 3.0 blue edition software (Zeiss, Jena, Germany).

The number of fluorescence-tagged LC3 puncta; the colocalization of lysosomal proteins and mitochondrial proteins with the fluorescence-tagged LC3 protein; the number of fluorescence-tagged LC3 puncta containing the fluorescence-tagged mitochondrial reporter; and the translocation of Parkin and TBK1 to mitochondria were analyzed with Zen 3.0 blue software and manually quantified. The number of mitolysosomes was semiautomatically quantified using the ImageJ Mito-QC counter plug-in following a procedure previously described [51]. To analyze mitophagic flux as indicated by MT-Keima reporter cells, the cells were directly observed by confocal microscopy as previously described [52]. Emission signals were obtained after excitation with a 488-nm laser (pH 4.0; shown in green) and a 561-nm laser (pH 7.0; shown in red). To assess the number of mitophagic cells with MT-Keima by FACS analysis, the cells were harvested, resuspended in PBS, and then analyzed by a Thermo Fisher Attune NxT Flow Cytometer (Thermo Fisher Scientific) using excitation wavelengths of 405 nm (pH 4.0) and 561 nm (pH 7.0) with 605/20 nm and 610/20 nm emission filters. A total of 30,000 events in each sample were collected and then assessed by FlowJo software (V10; Tree star). The quantification data were graphed using GraphPad Prism 5.0 software, and the statistical significance was assessed by two-tailed *t*-test with a confidence interval of 95% (* *p* < 0.05; ** *p* < 0.01; and *** *p* < 0.001).

### 2.6. Time-Lapse Live-Cell Imaging

Fluorescence-tagged reporter cells were plated at a density of 2 × 10^5^ cells on 35-mm high ibiTreat μ-Dishes (ibidi, Gräfelfing, Germany). Eight hours later, 0.2 μM baicalein was added to the cell culture and incubated for 0.5 h at 37 °C with 5% CO_2_ in an enclosed incubator chamber equipped with an LSM780 laser scanning confocal microscope. The cells were continuously maintained in the presence of baicalein in the same incubation system until live imaging was performed. The movies were generated by a time-lapse capture of images at 1 min intervals for a total of 2~3 h with a Plan-Apochromat 63×/1.4 Oil DIC M27 objective confocal microscope. The movies and serial frames of each movie were analyzed with Zen 3.0 blue edition software.

### 2.7. Transmission Electron Microscopy (TEM)

For conventional TEM analysis, the cells were harvested and washed with PBS three times, followed by 0.1 M CAB buffer, pH 7.2 twice. Subsequently, the cells were treated with fixative I (2.5% GA and 4% PFA in 0.1 M CAB buffer, pH 7.2) at RT for 2 h. After washing with 0.1 M CAB buffer three times (each time for 5 min), the cells were incubated with fixative II (1% OsO_4_ in 0.1 M CAB pH 7.2, 15 mg/mL K_4_[Fe(CN)_6_]) at RT for 1 h, followed by two washes with 0.1 M CAB buffer and three washes with double-distilled water (ddH_2_O). Then, the cells were incubated with 0.4% uranyl acetate at 4 °C for 1 h. After extensive washing with ddH_2_O, the cells were dehydrated in a graded series of ethanol and embedded in Spurr’s resin according to the manufacturer’s instructions. The embedded samples were trimmed and sectioned to 70 nm thickness by an EM UC7 ultramicrotome (Leica, Wetzlar, Germany). The cell sections were stained with 0.4% uranyl acetate at RT for 20 min and then incubated with 4% lead citrate solution for 6 min. Electron micrographs were obtained with a JEM 1230 transmission electron microscope (magnification, ×5000~10,000; 100 kV; Japan).

### 2.8. Correlative Light and Electron Microscopy (CLEM)

CLEM was performed according to previously described procedures [53,54,55]. The cells were plated at a density of 5 × 10^4^ onto poly-D-lysine-coated, 35-mm gridded glass-bottom cell culture dishes (MatTek P35G-1.5-14-CGRD) 12 h prior to baicalein treatment. The baicalein-treated cells were washed once with PBS and once with 0.1 M CAB buffer, pH 7.2; then, the nuclei were stained with Hoechst 33342. After three washes with 0.1 M CAB buffer, pH 7.2, the cells were incubated with fixative III (0.5% GA and 4% PFA in 0.1 M CAB buffer, pH 7.2) as described above at 37 °C for 30 min. Then, the cells were washed twice with 0.1 M CAB buffer, pH 7.2, and stored in 2 mL 0.1 M CAB buffer, pH 7.2. The images of fluorescence-tagged reporters in cells were obtained by a Plan-Apochromat 63×/1.4 Oil DIC M27 objective equipped with confocal microscopy. The cell of interest was first marked by grids on dishes; then, the images of different Z-positions (~20 stacks, one stack of 200 nm thickness) of a defined cell were captured to further correlate fluorescence images with electron micrographs. Subsequently, the cells were fixed with fixative II, washed, incubated with 0.4% uranyl acetate, dehydrated, and embedded in Spurr’s resin as described above. The embedded samples were then trimmed and sectioned by the EM UC7 ultramicrotome. Serial sections (~20 sections, one section of 70 nm) were collected and stained. Images of each section were captured by a JEM 1230 electron microscope. The EM images were assembled into one montage and used to relocate the position of the defined cell on confocal fluorescence Z-stack images. Finally, the confocal fluorescence images were aligned with the montage of EM images by Adobe Photoshop CS6 software.

### 2.9. Generation of Knockout Cells Using Clustered Regularly Interspaced Short Palindromic Repeat (CRISPR)/CRISPR-Associated Protein 9 (Cas9) Gene Editing

The generation of gene knockout cells was performed as previously described [56,57,58]. The CRISPR-guided RNAs targeted to genes of interest were designed and synthesized according to the gRNA sequences of the Human CRISPR Knockout Pooled Library [57] and the Human Activity-Optimized CRISPR Knockout Library [58]. The synthesized gRNA duplexes were subcloned into lentiCRISPR v2 (#52961, Addgene), a lentiviral vector coexpressing Cas9 and gRNA, allowing the selection of transduced cells by puromycin. Then, lentiCRISPR v2 containing the target gRNA was cotransfected with psPAX2 and pMD2.G into 293T cells to generate infectious lentiviruses expressing Cas9 and gRNA. The collection and transduction of lentiviruses were performed as described above. After transduction, the cells were incubated with complete medium containing 2 μg/mL puromycin for 14 days. The culture medium was removed and replenished with fresh medium supplemented with puromycin at a concentration of 2 μg/mL once every 2 days. Finally, the selected cells were pooled, expanded, and verified for the efficiency of gene knockout by SDS-PAGE and Western blotting. The sequences of targeted gRNAs are shown in Appendix A.

## 3. Results

### 3.1. Baicalein Induces the Formation of Autophagic Vacuoles

An increasing number of studies have suggested that baicalein may regulate autophagy in liver cancer cells to promote apoptosis and enhance chemosensitivity [44,47]. In addition, baicalein has also been shown to protect liver cells from damage through autophagy [38,48]. However, how baicalein regulates hepatic autophagy is still unknown. In this study, we investigated the biological activity of baicalein in the regulation of cellular autophagy and delineated the functional role of baicalein-regulated autophagy in cellular homeostasis. First, we found that the level of the PE-conjugated form of LC3B (also known as LC3B-II), a specific marker used for monitoring autophagosome maturation [59], was increased in a concentration-dependent manner in baicalein-treated Huh7 cells (Figure 1A, left panel). Similarly, baicalein upregulated LC3B-II expression in treated Huh7.5 and HepG2 cells (Appendix A, top row). In addition, prolonged baicalein treatment enhanced the level of LC3B-II in Huh7 cells (Figure 1A, right panel), as well as in Huh7.5 and HepG2 cells (Appendix A, bottom row). These results suggested that baicalein induced autophagosome formation in these cells. Next, we examined whether baicalein triggers autophagic vacuole biogenesis by analyzing the formation of fluorescence-tagged LC3-labeled puncta, a conventional method of assessing autophagic vacuole maturation [59,60]. As shown in Figure 1B, baicalein treatment significantly increased the number of RFP-LC3- and GFP-LC3-labeled puncta in Huh7 cells (left and right panels, respectively). Moreover, a time-lapse live-cell imaging analysis revealed the induction of RFP-LC3- and GFP-LC3-labeled autophagic vacuole formation in baicalein-treated Huh7 cells (Figure 1C, white arrowheads, and Appendix A). A TEM-based ultrastructural analysis was performed to characterize the morphogenesis of autophagic vacuoles in baicalein-treated cells [61,62]. We found that baicalein induced the initial formation of autophagic vacuoles (AVi), which were double-membraned structures in which organelles, such as mitochondria, were observed to be engulfed (Figure 1D, top row). Additionally, late-stage autophagic vacuoles encompassing a large portion of the interior material destined for degradation (AVd) were also detected in the treated cells (Figure 1D, top row). In sharp contrast, no detectable AVi or AVd was found in the untreated cells (Figure 1D, bottom row). Collectively, these results suggested that baicalein promoted autophagic vacuole biogenesis.

### 3.2. Baicalein Activates Complete Autophagy throughout Autolysosome Maturation

To verify whether baicalein-induced autophagy proceeds to autophagosome–lysosome fusion, we analyzed the colocalization of autophagic vacuoles and lysosomes in baicalein-treated cells. As shown in Figure 2A, we found that baicalein treatment induced dramatic colocalization of LAMP1, a lysosome-associated protein with RFP-LC3-labeled autophagic vacuoles, compared with that in untreated cells, in which no significant overlapping fluorescence signals were detected. In a similar fashion, another lysosomal protein, LAMP2, significantly colocalized with RFP-LC3-labeled autophagic vacuoles (Appendix A), suggesting that baicalein-activated autophagic vacuoles may fuse with lysosomes. Moreover, a tandem fluorescence reporter, RFP-GFP-LC3, was used to discriminate between neutral autophagosomes and acidic autolysosomes [60]. When autophagosomes fuse with lysosomes, the fluorescence of GFP in the RFP-GFP-LC3 reporter is quenched because the lysosomal environment is acidic (low pH), but RFP fluorescence remains stable. Using this strategy, we found that the numbers of both autophagosomes (RFP^+^/GFP^+^ puncta of RFP-GFP-LC3) and autolysosomes (RFP^+^/GFP^−^ puncta of RFP-GFP-LC3) were dramatically increased in baicalein-treated cells, whereas no significant puncta were detected in untreated cells (Figure 2B). As shown in Figure 2C and Appendix A, live-cell imaging specifically demonstrated the dynamic processes of autophagosome induction (indicated by white arrowheads) and autophagosome fusion with lysosomes (indicated by white arrows) in baicalein-treated cells. Moreover, baicalein treatment significantly increased the colocalization of RFP^+^/GFP^−^ puncta of the RFP-GFP-LC3 reporter with LAMP1 and LAMP2 (Appendix A, respectively), confirming autophagosome–lysosome fusion in baicalein-treated cells. Furthermore, to determine whether baicalein enhances autophagic flux, the effect of BAF-A1, a vacuolar ATPase inhibitor that suppresses autolysosome maturation, was assessed [62,63]. As shown in Figure 2D, treatment with BAF-A1 not only further induced the expression of LC3B-II in baicalein-treated cells, but also led to greater LC3B-II accumulation in baicalein-treated cells than in untreated cells, suggesting that baicalein increased autophagic flux. Together, these experiments suggested that baicalein induced autophagy through the autolysosome maturation stage and that sequestrated material was degraded by lysosomal proteases.

### 3.3. Baicalein Induces Autophagy through the Ubiquitin-Like (UBL) Conjugation System

Next, we examined whether UBL conjugation in autophagosome maturation is required for baicalein-induced autophagy. A CRISPR/Cas9 gene editing strategy was used to knock out the expression of ATGs involved in UBL conjugation. ATG3 knockout (KO), ATG5KO, ATG7KO, and ATG16KO cells were successfully generated (Appendix A), treated with baicalein, and tested for LC3B-II induction (Figure 3A). Baicalein-induced LC3B-II expression was almost completely suppressed in ATG5KO and ATG7KO cells compared with parental cells (Figure 3A). Compared with parental cells, LC3B-II expression was apparently reduced in baicalein-treated ATG3KO and ATG16KO cells, but a small amount of LC3B-II was still detectable (Figure 3A). Consistent with the repression of LC3B-II expression, the baicalein-triggered formation of RFP-LC3-labeled autophagic vacuoles in these ATGKO cells was significantly inhibited compared with that in parental cells (Figure 3B). Similarly, using the RFP-GFP-LC3 reporter, the formation of RFP^+^/GFP^+^ autophagosomes and RFP^+^/GFP^−^ autolysosomes was largely diminished in these KO cells (Appendix A). These results collectively indicated that baicalein activated complete autophagy through UBL conjugation-dependent autophagosome maturation.

### 3.4. Baicalein Activates the Autophagic Process to Engulf Mitochondria

Mounting evidence has indicated that autophagy may selectively target mitochondria for degradation through mitophagy [16,17,18]. Given that our TEM analysis specifically showed that mitochondria were sequestered within AVi in baicalein-treated cells (Figure 1D, top row), we then investigated whether baicalein-induced autophagy participates in mitochondrial turnover. As shown in Appendix A, significant colocalization between HSP60, a mitochondrial chaperonin protein, and RFP-LC3 puncta in baicalein-treated cells was observed (indicated by white arrowheads in magnified field-1); however, no detectable colocalized signal was observed in untreated cells (Appendix A). In addition, increased colocalization of TOM20 with baicalein-induced RFP-LC3 puncta was detected, as shown in Figure 4A (indicated by white arrowheads in magnified field-1). Analogously, the GFP-LC3-labeled autophagic vacuoles colocalized with HSP60 and TOM20 in baicalein-treated cells (Appendix A and Figure 4B, respectively; indicated by white arrowheads in magnified field-1). Moreover, significant colocalization between RFP^+^/GFP^−^ autolysosomes in the RFP-GFP-LC3 reporter and HSP60 was detected in baicalein-treated cells (Figure 4C; indicated by white arrowheads in magnified field-1). Collectively, these results suggested that baicalein-induced autophagy may engulf mitochondria.

Next, using the Mito-GFP reporter, a reporter of the mitochondrial targeting sequence (MTS) in human cytochrome c subunit VIII oxidase (COX8A) fused with GFP, we found that baicalein triggered the fragmentation of mitochondria, as demonstrated by the large population of fission mitochondria in baicalein-treated cells (Appendix A). In contrast, a majority of mitochondria remained elongated and fused in untreated cells (Appendix A). In addition, the aggregate of deformed mitochondria was detected in the EM analysis of baicalein-treated cells, compared with the dispersed distribution of elongated tubular mitochondria in untreated cells (Appendix A). These results implied that baicalein induced mitochondrial fragmentation and clustering, a characteristic of mitochondrial autophagy. Moreover, the sequestration of mitochondria within the autophagic process was specifically traced in cells simultaneously expressing RFP-LC3 and Mito-GFP. As shown in Figure 5A, the colocalization between RFP-LC3-labeled autophagic vacuoles and Mito-GFP-marked mitochondria was significantly higher in baicalein-treated cells than in untreated cells (indicated by white arrowheads in magnified field-1). Time-lapse live-cell imaging demonstrated the engulfment of Mito-GFP within the RFP-LC3-labeled autophagic vacuoles in baicalein-treated cells (Figure 5B and Appendix A). In a similar fashion, baicalein treatment promoted the engulfment of Mito-RFP, a mitochondrial reporter of the COX8A MTS fused with RFP, by GFP-LC3-labeled autophagic vacuoles (Appendix A; indicated by white arrowheads in magnified field-1). The dynamic process of Mito-RFP sequestration by GFP-LC3-labeled autophagic vacuoles in baicalein-treated cells was confirmed by live-cell imaging, as shown in Appendix A. Moreover, we found a dramatic increase in the overlapping signals of RFP^+^/GFP^−^ autolysosomes in the RFP-GFP-LC3 reporter with those of Mito-miRFP670, a miRFP670-tagged mitochondrial reporter, in cells treated with baicalein (Appendix A; indicated by white arrowheads in magnified field-1). Additionally, time-lapse live-cell imaging delineated the sequential engulfment process of mitochondria by RFP^+^/GFP^+^ autophagosomes (indicated by white arrowheads) and RFP^+^/GFP^−^ autolysosomes (indicated by white arrows) in baicalein-treated cells (Appendix A). Therefore, these studies implied that autophagy in baicalein-treated cells led to mitochondrial sequestration for subsequent degradation.

Furthermore, we used CLEM to analyze mitochondrial engulfment at the ultrastructural level during baicalein-induced autophagy. In baicalein-treated RFP-LC3/Mito-GFP cells, confocal microscopy images taken at different Z-positions (Figure 6A) were assembled to reconstitute the 3-D deconvoluted structure shown in Figure 6B. In this 3-D reconstituted structure, a perinuclear area with RFP-LC3 puncta encompassing Mito-GFP was detected (Figure 6B, white dashed box). After aligning the confocal microscopy and EM images, the CLEM results shown in Figure 6C revealed that RFP-LC3 puncta overlapping the Mito-GFP area (IF image in right middle panel) contained autophagic vacuoles (EM image in right bottom panel) in which numerous degradative mitochondria had been engulfed (arrowheads in the EM image), directly indicating that baicalein-induced autophagy participated in the elimination of mitochondria. Taken together, these results indicated that baicalein activated autophagy to promote mitochondrial turnover.

### 3.5. Activation of Mitophagy by Baicalein

To confirm that baicalein-induced autophagy targets mitochondria for degradation, the Mito-QC reporter, a tandem RFP-GFP tag fused with MTS (amino acids 101–152) in human fission 1 (FIS1), was used to assess mitochondrial turnover in baicalein-treated cells. When mitochondria are targeted to autolysosomes for degradation, the GFP signal emitted by the Mito-QC reporter is quenched by the low lysosomal pH, whereas the RFP signal emitted by Mito-QC remains stable. This reporter has been used to quantitatively assess mitophagy in vivo and in vitro [64,65], and using it, we found that baicalein significantly triggered the formation of mitolysosomes (RFP^+^/GFP^−^ dot-like structures; indicated by white arrowheads in magnified field-1), but the dominant RFP^+^/GFP^+^ signal of Mito-QC remained dominant in the untreated cells (Figure 7A). In addition, the RFP^+^/GFP^−^ mitolysosomes colocalized with LAMP1 and LAMP2 (Appendix A, respectively; indicated by white arrowheads in magnified field-1). A live-cell imaging study illustrated the formation of mitolysosomes in baicalein-treated cells (white arrowheads in Figure 7B and Appendix A). In addition, the MT-Keima reporter, the human FIS1 MTS fused with Keima (a dual-excitation fluorescence protein that undergoes a shift from a shorter to a longer wavelength in the lysosomal acidic environment), was employed to monitor baicalein-activated mitophagic flux. In the physiological pH environment, the excitation of MT-Keima at the short wavelength (488 nm, green) was predominant; then, the MT-Keima emissions showed excitation at the long wavelength (561 nm, red), indicating that the reporter had been exposed to the lysosome acidic environment because of active mitophagy. Notably, the MT-Keima reporter allowed us to feasibly measure mitophagic flux with higher sensitivity than was possible with Mito-QC [52,66]. As shown by the emissions data presented in Appendix A, baicalein treatment dramatically induced the excitation of MT-Keima at the long wavelength (left panel), but no significant emission indicating excitation of MT-Keima at the long excitation wavelength was visible in untreated cells (right panel). These results suggested that baicalein treatment increased the number of mitophagic cells with acidic MT-Keima. Additionally, FACS analysis confirmed that baicalein treatment induced an increase in cells containing acidic MT-Keima, in a similar fashion to that of cells treated with CCCP, a common inducer used in studying mitophagy [67,68] (Appendix A). In contrast, no significant cell population of acidic MT-Keima was induced by nutrient starvation (Appendix A). Notably, we found that the quantification of acidic MT-Keima cells using confocal microscopy achieved superior sensitivity than that analyzed by FACS analysis (as demonstrated by comparison of the % cells with acidic MT-Keima between Appendix A). In addition, the time-lapse live-cell imaging analysis revealed the emergence of an MT-Keima signal excited by longer wavelengths in baicalein-treated cells (white arrowheads in Appendix A), suggesting that baicalein increased mitophagic flux. Furthermore, the time-dependent degradation of mitochondria-associated proteins, including HSP60, TOM20, and COX IV, in baicalein-treated cells directly showed that baicalein induced mitochondrial degradation (Figure 7C). In conclusion, these results indicated that baicalein activated mitophagy to degrade mitochondria.

### 3.6. Baicalein Induces the Translocation of Parkin to Mitochondria, the Phosphorylation of Ubiquitin at Ser65, and the Stabilization of PINK1

Next, we tested whether baicalein treatment triggers the translocation of Parkin to mitochondria, indicating the initiation of PINK1/Parkin-dependent mitophagy [22]. As shown in Figure 8A, a significant proportion of RFP-Parkin was specifically translocated to areas marked by Mito-GFP in baicalein-treated cells (indicated by white arrowheads in magnified field-1). Nearly 80% of baicalein-treated cells revealed mitochondrial translocation of Mito-GFP (Figure 8A, right panel). Time-lapse live-cell imaging also revealed the baicalein-triggered dynamic translocation of RFP-Parkin to Mito-GFP areas (white arrowheads in Figure 8B and Appendix A). Consistent with this finding, baicalein induced YFP-Parkin translocation to Mito-RFP-labeled areas (Appendix A; indicated by white arrowheads in magnified field-1). The mitochondrial translocation of YFP-Parkin to a Mito-RFP labeled area was verified by live-cell imaging data presented in Appendix A (indicated by white arrowheads) and Appendix A. These results together implied that baicalein induced Parkin translocation to mitochondria. Moreover, we demonstrated that miRFP670-LC3-labeled autophagic vacuoles were specifically recruited to mitochondria where RFP-Parkin had been translocated (white arrowheads in Appendix A), suggesting that mitochondrial translocation of Parkin coincides with the formation of autophagosomes in baicalein-activated mitophagy. As previous studies have shown that PINK1 phosphorylates ubiquitin at Ser65 to activate Parkin during mitophagy [20,21], we next investigated whether ubiquitin is phosphorylated at Ser65 in baicalein-induced mitophagy. As shown in Figure 8C, an increased population of phospho-ubiquitin (Ser65) was detected in the colocalized area of RFP-LC3-labeled autophagic vacuoles and Mito-GFP in baicalein-treated cells (indicated by white arrowheads in magnified field-1). Similarly, baicalein treatment also triggered the expression of Ser65-phosphorylated ubiquitin on mitochondria after RFP-Parkin translocation (Figure 8D; indicated by white arrowheads in magnified field-1). Together, these results suggested that baicalein activated mitophagy through the translocation of Parkin to mitochondria and the phosphorylation of ubiquitin at Ser65. Previous studies have shown that Parkin-dependent mitophagy promotes the ubiquitination of MFN1 and MFN2 and leads them to be degraded in a proteasome- and p97-depedent manner [69,70]. On the other hand, the stabilization of PINK1 on depolarized mitochondria was demonstrated to activate Parkin in mitophagy [67,68]. In accordance with these studies, we found that baicalein treatment led to MFN1 and MFN2 degradation in a concentration-dependent manner compared with untreated cells (Figure 8E). Similarly, CCCP also resulted in the proteolysis of MFN1 and MFN2, whereas nutrient starvation had no significant effect on MFN1 and MFN2 expression (Figure 8E). Intriguingly, ectopic overexpression of mTagBFP2-Parkin did not further increase the baicalein- and CCCP-induced MFN1 and MFN2 degradation (Figure 8E), presumably due to the abundant expression of endogenous Parkin in Huh7 cells. In addition, we found that baicalein and CCCP elevated the expression of PINK1, including precursor and mature forms of PINK1, in the purified mitochondria of treated cells (Figure 8F, left panel). Furthermore, MFN2 polyubiquitination was significantly detected in the mitochondrial fraction of baicalein- and CCCP-treated cells (Figure 8F, right panel). Collectively, these results showed that baicalein induced Parkin-dependent mitophagy by stabilizing PINK1 in mitochondria and promoting MFN2 polyubiquitination.

### 3.7. Translocation of TBK1 to Mitochondria and Phosphorylation of TBK1 at Ser172 in Baicalein-Treated Cells

The translocation of TBK1 to mitochondria and the TBK1-mediated phosphorylation of p62/SQSTM1 and OPTN have been implicated in PINK1/Parkin-dependent mitophagy [24,25,26]. We therefore assessed whether baicalein induces mitophagy through TBK1 translocation to mitochondria. As shown in Figure 9A, significant colocalization of GFP-TBK1 with RFP-LC3 puncta was detected in baicalein-treated cells (indicated by white arrowheads in magnified field-1), suggesting that TBK1 might have been involved in baicalein-induced autophagic processes. In addition, we found that baicalein induced high levels of GFP-TBK1 translocation to Mito-miRFP670-marked mitochondria (Appendix A; indicated by white arrowheads in magnified field-1). Moreover, a significant proportion of Mito-miRFP670 was sequestered in an area with overlapping RFP-LC3 puncta and GFP-TBK1 signals in baicalein-treated cells (Figure 9B; indicated by white arrowheads in magnified field-1), suggesting that TBK1 might have participated in baicalein-activated mitophagy. The time-lapse live-cell imaging specifically indicated a coordinated mechanism involving GFP-TBK1 in the engulfment of Mito-miRFP670 by RFP-LC3-labeled autophagic vacuoles in baicalein-treated cells (white arrowheads in Appendix A), directly implying that baicalein induced TBK1 translocation during mitophagy. Furthermore, baicalein treatment activated TBK1 phosphorylation at Ser172 (Figure 9C). The baicalein-induced TBK1 Ser172 phosphorylation coincided with the degradation of MTCO2, which was shown to monitor PINK1/Parkin-dependent mitophagy [23]. In particular, Ser172-phosphorylated TBK1 was shown to localize to mitophagic vacuoles, as indicated by the merging of the Mito-GFP signal with RFP-LC3 puncta in baicalein-treated cells (Figure 9D; indicated by white arrowheads in magnified field-1). Collectively, these results indicated that baicalein induced TBK1 phosphorylation at Ser172 and drove the translocation of TBK1 to mitochondria during mitophagy.

### 3.8. Baicalein Activates the Recruitment of NDP52 and OPTN in Mitophagy

Several studies have identified active cargo receptors in PINK1/Parkin-dependent mitophagy [27,28]. Therefore, we tested known mitophagy receptors, namely, SQSTM1, NBR1, NDP52, OPTN, TAX1BP1, and BINP3L, to determine whether they each specifically participated in baicalein-activated mitophagy. Through confocal microscopy analysis, we found that, among the aforementioned cargo receptors, GFP-NDP52 and GFP-OPTN specifically colocalized with RFP-LC3 puncta that had formed after baicalein treatment (Figure 10, left bottom and right top panels; indicated by white arrowheads in magnified field-1). In contrast, no significant SQSTM1, NBR1, TAX1BP1, or BNIP3L signals overlapped with RFP-LC3 puncta in baicalein-treated cells (Figure 10). These results suggested that NDP52 and OPTN might have participated in baicalein-activated autophagy. In addition, the simultaneous localization of Mito-miRFP670 and GFP-NDP52 or GFP-OPTN, but not other cargo receptors, was drastically increased in baicalein-treated cells (Figure 11, left bottom and right top panels; indicated by white arrowheads in magnified field-1). In contrast, no detectable GFP-NDP52 or GFP-OPTN signals were observed in areas labeled with Mito-miRFP670 in untreated cells (Figure 11). These results strongly suggested that baicalein induced the translocation of NDP52 and OPTN to mitochondria. Most importantly, the signals emitted by translocated GFP-NDP52 and GFP-OPTN (the GFP-NDP52 or GFP-OPTN signals merged with Mito-miRFP670) were associated with RFP-LC3 puncta in baicalein-treated cells (Figure 12A, left bottom and right top panels; indicated by white arrowheads in magnified field-1), implying that the mitochondrial recruitment of NDP52 and OPTN might have been involved in baicalein-induced mitophagy. Time-lapse live-cell imaging revealed that GFP-NDP52 and GFP-OPTN were involved in the elimination of Mito-miRFP670-marked mitochondria by RFP-LC3-labeled autophagic vacuoles (white arrowheads in Figure 12B,C; Appendix A, respectively). Together, these data indicated that NDP52 and OPTN might have functioned in mitochondrial turnover during mitophagy in baicalein-treated cells.

### 3.9. Activation of Mitophagy Induced by NDP52 and OPTN in Baicalein-Treated Cells

We then employed CLEM analysis of the cellular ultrastructure to determine whether NDP52 and OPTN participate in the elimination of mitochondria through baicalein-activated mitophagy. In baicalein-treated cells, deformed mitochondria were located inside RFP-LC3-labeled autophagic vacuoles with overlapping GFP-NDP52 signals (Appendix A). Similarly, GFP-OPTN-associated RFP-LC3-labeled autophagic vacuoles were shown to have sequestered numerous degrading mitochondria in baicalein-treated cells (Appendix A). These results together indicated that NDP52 and OPTN were involved in the degradation of mitochondria during baicalein-induced mitophagy. Finally, we examined whether interference with autophagosome maturation, achieved by knocking out the expression of ATGs involved with UBL conjugation, inhibits mitochondrial turnover in baicalein-treated cells. As shown in Figure 13A,B, we found that the sequestration of Mito-GFP within RFP-LC3-labeled autophagic vacuoles and the level of mitophagic flux measured by Mito-QC were significantly reduced in baicalein-treated ATG3KO, ATG5KO, ATG7KO, and ATG16KO cells compared with those in parental cells treated with baicalein. By confocal and FACS analyses, we further demonstrated that the baicalein-induced increase in the number of mitophagic cells containing acidic MT-Keima was drastically diminished in ATG3KO, ATG5KO, ATG7KO, and ATG16KO cells compared with parental cells (Appendix A). These results implied that baicalein-activated autophagy played a critical role in the elimination of mitochondria. In addition, we examined whether knocking out the expression of mitophagy receptor genes suppresses baicalein-induced mitochondrial degradation through mitophagy. SQSTM1, NDP52, OPTN, TAX1BP1, or BNIP3L expression was successfully knocked out in individual cell lines (Appendix A). Baicalein failed to trigger Mito-GFP engulfment by RFP-LC3-labeled autophagic vacuoles in the NDP52KO and OPTNKO cell lines (Figure 13C). Additionally, mitophagic flux, as measured by the Mito-QC assay, and the number of mitophagic cells labeled by acidic MT-Keima were markedly reduced in the NDP52KO and OPTNKO cells treated with baicalein (Figure 13D and Appendix A, respectively). In sharp contrast, the SQSTM1KO, TAX1BP1KO, and BNIP3LKO cell lines showed a degree of baicalein-induced sequestration of Mito-GFP within RFP-LC3-labeled autophagic vacuoles similar to that of parental cells (Figure 13C); an amount of baicalein-enhanced mitophagic flux, as measured by Mito-QC analysis (Figure 13D); and a number of mitophagic cells labeled by acidic MT-Keima (Appendix A) comparable with that of parental cells. Collectively, these results indicated that baicalein activated NDP52- and OPTN-dependent mitophagy, leading to mitochondrial turnover.

## 4. Discussion

In the past decade, accumulating evidence has shown that diverse mechanisms underlie the regulation of mitophagy [16,17]. Additionally, several mitophagy-related receptors have been identified [17,19,20,21,22,23]. Most importantly, the dysregulation of mitophagy has been implicated in the development of human diseases, including liver diseases [29,30,31]. Therefore, a comprehensive understanding of mitophagy regulation may lead to the development of potential strategies to treat human diseases [32,71,72]. Notably, the basis for the current understanding of mitophagy activation, particularly for PINK1/Parkin-dependent mitophagy, is an experimental model of ectopically Parkin-expressing HeLa cells lacking endogenous Parkin and induced with extreme stimuli, such as uncoupling agents [17,19,20,21,22,23]. Therefore, whether PINK1/Parkin-dependent mitophagy is regulated in cells that autonomously express Parkin, such as liver cells, in the same way as the HeLa model suggests is still elusive. In addition, whether PINK1/Parkin-dependent mitophagy is analogously activated under physiological conditions to promote mitochondrial turnover, so-called basal mitophagy, is even less clear, and the mechanism is unknown [28,73].

On the other hand, interference with autophagy in genetic mouse models and human patients has been demonstrated in the pathogeneses of liver diseases, such as liver cancer [73,74,75,76]. Genetic studies have shown that the liver-specific knockout of ATG5 and ATG7 expression induces the spontaneous development of HCC, in which numerous deformed mitochondria and ubiquitin-containing compartments are observed [73,74]. In addition to that in liver cancer, the accumulation of damaged mitochondria by impaired mitophagy is involved in the development of fatty liver disease [77,78]. Therefore, strategies for enhancing hepatic autophagy and/or mitophagy to treat liver diseases can be rationally developed [32,71,72]. Currently, small molecules for regulating hepatic mitophagy, such as mitophagy inducers that can be used in the clinic, are unavailable, probably due to limited knowledge of mitophagy regulation in liver cells. Therefore, comprehensive delineation of the mechanism by which hepatic mitophagy regulates mitochondrial turnover is urgently needed.

Baicalein is a flavonoid isolated from the Chinese herb *S. baicalensis* that has hepatoprotective activity and has been used for a long time as a traditional Chinese medicine [33,34,37,38]. However, the mechanisms by which baicalein promotes the maintenance of liver physiology and protects against liver damage are unclear. In this study, we provided compelling evidence showing that baicalein activates hepatic autophagy and induces mitophagy to promote mitochondrial turnover. Through IF/confocal microscopy, time-lapse live-cell imaging, and TEM-based ultrastructural analyses (Figure 1, Figure 2 and Figure 3 and Appendix A), we demonstrated that baicalein induces the entire autophagic process and increases autophagic flux. Through integrated results produced by cell imaging technologies, including confocal microscopy, time-lapse live imaging, and CLEM, we showed details of the dynamics of baicalein-induced mitophagy (Figure 4, Figure 5 and Figure 6 and Appendix A). Using Mito-QC and MT-Keima reporter assays, we provided convincing evidence that baicalein promotes the formation of mitolysosomes and increases mitophagic flux, eventually promoting mitochondrial degradation (Figure 7 and Appendix A). Moreover, our study mechanistically uncovered the concerted action of regulators of mitophagy activation in baicalein-treated cells, including the mitochondrial translocation of Parkin and TBK1, PINK1 the stabilization in mitochondria, and the phosphorylation of ubiquitin at Ser65 and TBK1 at Ser172 (Figure 8 and Figure 9, and Appendix A). Furthermore, our findings indicated that NDP52 and OPTN are specific cargo receptors in Parkin-mediated mitophagy in baicalein-treated cells (Figure 10, Figure 11 and Figure 12 and Appendix A), in line with a previous study reporting that these two cargo receptors function in PINK1/Parkin-dependent mitophagy in HeLa cells [17,23]. Finally, the suppression of baicalein-induced mitochondrial degradation by knocking out the gene expression of ATGs involved in the formation of autophagosomes and the reduction in mitochondrial turnover in baicalein-treated NDP52KO and OPTNKO cells (Figure 13 and Appendix A) revealed the crucial roles played by these ATGs in autophagosome maturation and the cargo receptors, NDP52 and OPTN, in baicalein-activated Parkin-dependent hepatic mitophagy. Taken together, these results provide the first line of evidence showing not only that baicalein induces Parkin-dependent mitophagy in liver cells, but also the details that delineate the molecular regulation of hepatic mitophagy.

Although the findings presented herein convincingly underscore that baicalein induces hepatic autophagy, the early events regulating the biogenesis of autophagosomes in baicalein-induced autophagy remain unclear. Previous studies have noted that baicalein represses the phosphorylation of mTOR and activates ULK1 signaling [79,80], suggesting that baicalein may activate hepatic autophagy by suppressing the activation of mTOR and/or ULK phosphorylation. Moreover, a recent study showed that baicalein triggered ER stress to protect the liver from CCl_4_-induced damage [48]. In accordance with these findings, whether ER stress and downstream unfolded protein response (UPR) signaling can trigger the initiation of baicalein-induced autophagy by transcriptionally controlling ATG gene expression in liver cells can be examined. On the other hand, the membrane source of IM/phagophore in baicalein-induced autophagy is also unknown, and further investigation is still needed. Interestingly, we detected a large portion of the ER membrane surrounding mitochondria in baicalein-treated cells (data not shown), suggesting that ER and mitochondria-associated ER membranes (MAMs) may supply membranous components for baicalein-induced autophagy and mitophagy. Notably, ER strands have recently been shown to provide a platform for mitophagy activation [81].

Previous studies have indicated that the regulators of the autophagy initiation complex, including ULK1, DFCP1, WIPI, and FIP200, are translocated to damaged mitochondria, thereby proximately triggering autophagosome maturation in mitophagy [23,81,82]. In accordance with these findings, the formation of miRFP670-LC3-labeled autophagic vacuoles in the turnover of mitochondria by Parkin-dependent mitophagy in baicalein-treated cells (Figure 8E) supports the notion that autophagy-related factors for initiating autophagosome formation may be locally recruited to mitochondria by baicalein during mitophagy activation. TBK1 is activated by autophosphorylation at Ser172, and TBK1-induced phosphorylation of cargo receptors, such as SQSTM1 and OPTN, have been implicated in mitophagy regulation [24,25,26]. In line with these observations, it is rational to ask whether TBK1 regulates baicalein-induced mitophagy by promoting the phosphorylation of NDP52 and OPTN in hepatic cells. Most importantly, whether mitochondria are deformed during mitophagy activation and the mechanism triggering mitochondrial depolarization in baicalein-treated cells remain unknown. Very recently, a study showed that in neuronal cells, 12/15-lipoxygenase (12/15-LOX) disrupts mitochondrial health, and baicalein treatment preserves the integrity of mitochondria by inhibiting 12/15-LOX activity [83]. Therefore, determining whether 12/15-LOX is involved in baicalein-induced hepatic mitophagy is an interesting research direction.

Can baicalein-induced autophagy and mitophagy be applied to treat liver diseases? The answer remains unclear because whether baicalein induces autophagy and mitophagy in normal liver cells and in vivo remains to be determined. Further investigations are urgently needed to verify the autophagy- and mitophagy-induction ability of baicalein in primary human hepatocytes (PHHs) and experimental mouse models of liver diseases. Intriguingly, recent studies have implied that baicalein suppresses the accumulation of hepatic fat and prevents the formation of fatty liver in mice [84,85]. Asking whether baicalein-induced autophagy and mitophagy are involved in these beneficial effects is a worthy inquiry. In contrast to the protective roles of mitophagy in liver cancer, the activation of mitophagy by CCCP in hepatic cells has been shown to promote the development of liver cancer by repressing p53 phosphorylation at Ser392 and maintaining the stemness of cancer stem cells (CSCs) [86], suggesting that the induction of hepatic mitophagy may not be an ideal treatment for liver cancer. Notably, we failed to detect baicalein suppression of p53 phosphorylation at Ser392 or baicalein induction of CSC-related gene expression, such as Nanog and Oct4 expression (data not shown), implying that baicalein-induced Parkin-dependent mitophagy is dispensable for p53 phosphorylation-dependent self-renewal of CSCs and the development of liver cancer. Nevertheless, we conclude that baicalein is a novel inducer of hepatic autophagy and mitophagy that can potentially be used to treat liver diseases.

## Figures and Tables

**Figure 1 cells-11-01132-f001:**
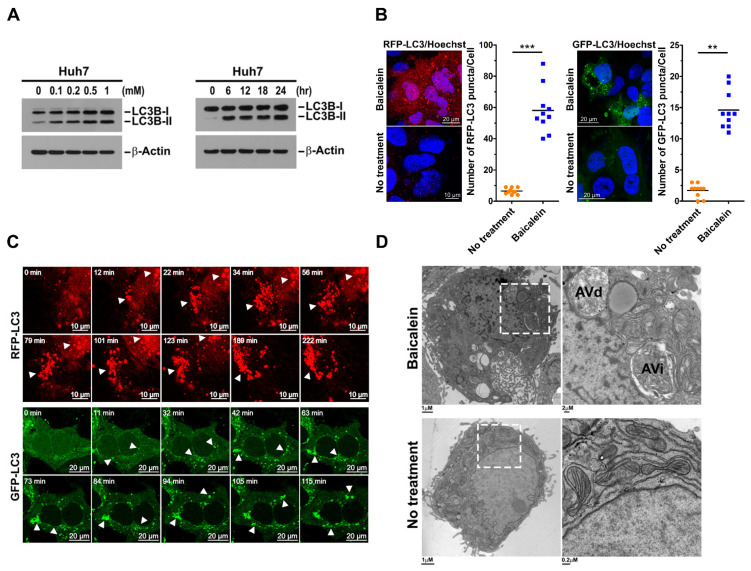
Formation of autophagic vacuoles induced by baicalein: (**A**) Left panel: Huh7 cells were treated with baicalein at the indicated concentrations. Twelve hours later, the cells were harvested and analyzed for protein expression by Western blotting. Right panel: Huh7 cells were treated with 0.5 mM baicalein and harvested at different times for use in the Western blot analysis. β-Actin was immunoblotted as the internal control used to ensure that equal amounts of protein in each sample were loaded. (**B**) Huh7 cells stably expressing RFP-LC3 and GFP-LC3 (Huh7/RFP-LC3 and Huh7/GFP-LC3 cells, respectively) were established by lentiviral gene delivery as described in “Materials and Methods”. Huh7/RFP-LC3 (left panel) and Huh7/GFP-LC3 cells (right panel) were treated with 0.5 mM baicalein for four hours and analyzed by confocal microscopy. Hoechst 33342 staining (blue) indicates the loci of nuclei. The number of RFP-LC3 and GFP-LC3 puncta was quantified. The data represent the means ± SEM (*n* = 10, *** *p* < 0.001, ** *p* < 0.01). (**C**) Selected live imaging frames showing Huh7/RFP-LC3 and Huh7/GFP-LC3 cells treated with 0.5 mM baicalein. RFP-LC3- and GFP-LC3-labeled autophagic vacuoles are shown by white arrowheads. Movies of the live imaging are presented in Appendix A. (**D**) The baicalein-treated Huh7 cells described in (**C**) were analyzed by TEM. The right panel shows the enlargement of the white dashed boxes presented in the left panels. AVi, initial-stage autophagic vacuole; AVd, late-stage and degradative autophagic vacuole.

**Figure 2 cells-11-01132-f002:**
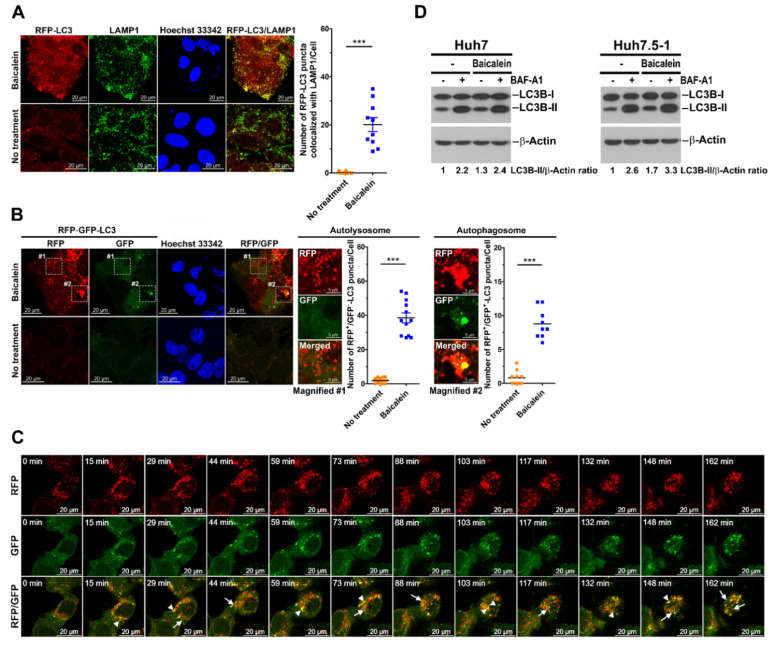
Activation of complete autophagy by baicalein: (**A**) Huh7/RFP-LC3 cells were treated with or without 0.5 mM baicalein. Four hours later, the cells were fixed and immunostained with LAMP1 (green) and Hoechst 33342 (blue). The number of RFP-LC3 puncta colocalized with LAMP1 was quantified. The data represent the means ± SEM (*n* = 10, *** *p* < 0.001). (**B**) Huh7 cells stably expressing RFP-GFP-LC3 (Huh7/RFP-GFP-LC3 cells) were established by lentiviral gene delivery as described in the “Materials and Methods” and then cultured in the presence or absence of 0.5 mM baicalein for four hours. Then, the cells were fixed and stained with Hoechst 33342 for nuclear identification. RFP^+^/GFP^−^ puncta (autolysosome) and RFP^+^/GFP^+^ puncta (autophagosome) are indicated in magnified field-1 and magnified field-2, respectively. White dashed boxes 1 and 2 show the areas of baicalein-treated cells at a higher magnification. The numbers of autophagosomes and autolysosomes were quantified, and the data represent the means ± SEM (*n* = 10, *** *p* < 0.001). (**C**) Selected live imaging frames showing 0.5 mM baicalein-treated Huh7/RFP-GFP-LC3 cells. The full-length video showing the live imaging is presented in Appendix A. The white arrowheads and arrows in a representative image indicate the formation of RFP^+^/GFP^+^ autophagosomes and RFP^+^/GFP^−^ autolysosomes, respectively. (**D**) Huh7 and Huh7.5 cells were cultured in the presence or absence of 0.2 mM baicalein for six hours. Then, the baicalein-treated cells and untreated cells were treated without (−) and with (+) 50 nM BAF-A1. Three hours later, the cells were harvested, and protein expression was analyzed by Western blotting. The relative expression ratio of LC3B-II to β-actin was determined by gel densitometry using ImageJ.

**Figure 3 cells-11-01132-f003:**
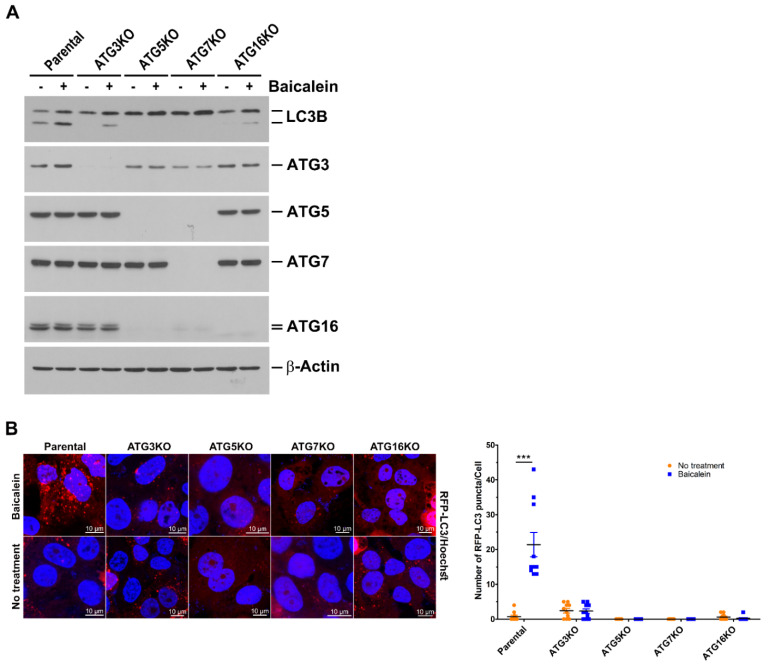
Dependence of the UBL conjugation system on baicalein-induced autophagy: (**A**) Individual ATG genes were knocked out in Huh7 cells by CRISPR/Cas9 gene editing as described in “Materials and Methods”. The parental and the individual ATG knockout (KO) cell lines were treated with or without 0.5 mM baicalein for six hours and harvested for analysis of protein expression by Western blotting. (**B**) Huh7 parental and individual ATG-KO cells stably expressing RFP-LC3 were established by lentiviral gene delivery and then treated with or without 0.5 mM baicalein. Four hours later, the cells were fixed, stained for nuclei with Hoechst 33342, and then analyzed by confocal microscopy. The number of RFP-LC3 puncta in parental and individual ATG-KO cell lines was quantified. The data represent the means ± SEM (*n* = 10, *** *p* < 0.001).

**Figure 4 cells-11-01132-f004:**
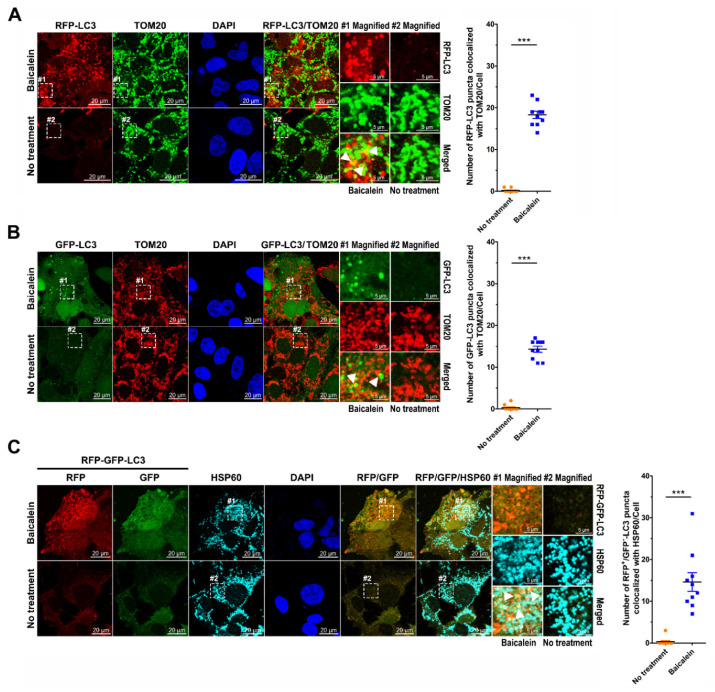
Colocalization of autophagic vacuoles and mitochondria in baicalein-treated cells: (**A**) Huh7/RFP-LC3 cells were treated with or without 0.5 mM baicalein for four hours. Then, the cells were fixed, immunostained for TOM20 (green) and nuclei by DAPI (blue), and analyzed by confocal microscopy. (**B**) Huh7/GFP-LC3 cells were treated with baicalein as described above, immunostained for TOM20 (green) and nuclei (blue), and assessed by confocal microscopy. (**C**) Huh7/RFP-GFP-LC3 cells were cultured in the presence or absence of 0.5 mM baicalein. Four hours later, the cells were fixed and immunostained for HSP60 (cyan) and DAPI (blue) and subjected to confocal microscopy analysis. The number of RFP-LC3 puncta colocalized with TOM20 (**A**), the number of GFP-LC3 puncta colocalized with TOM20 (**B**), and the number of RFP^+^/GFP^−^ autolysosomes overlapping with the HSP60 signal (**C**) were quantified. The data represent the means ± SEM (*n* = 10, *** *p* < 0.001). In each panel, magnified field-1 and magnified field-2 show enlarged images of the areas in white dashed boxes 1 and 2 in the images of baicalein-treated and untreated cells, respectively. The white arrowheads in magnified field-1 indicate the colocalized signal.

**Figure 5 cells-11-01132-f005:**
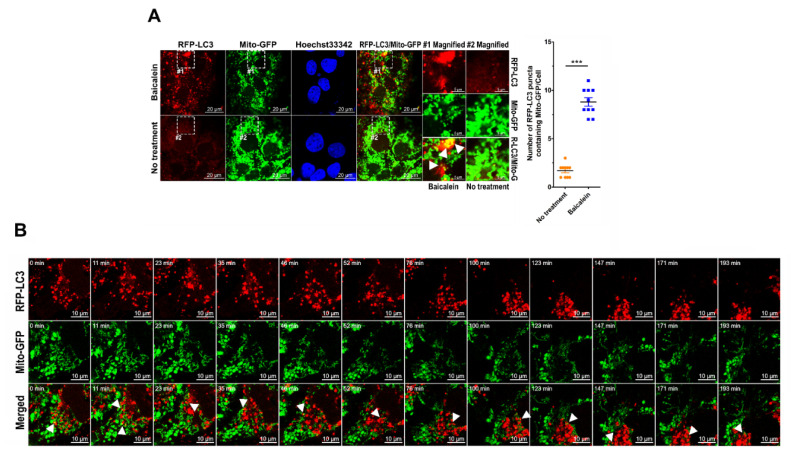
Dynamics of mitochondrial sequestration in baicalein-induced autophagy: (**A**) Huh7 cells stably coexpressing RFP-LC3 and Mito-GFP (Huh7/RFP-LC3/Mito-GFP) were established by lentiviral gene delivery as described in “Materials and Methods”. Huh7/RFP-LC3/Mito-GFP cells were treated with or without 0.5 mM baicalein for four hours, fixed and stained with Hoechst 33342, and analyzed by confocal microscopy. The degree of colocalization between Mito-GFP-labeled mitochondria and RFP-LC3 puncta was quantified. The data represent the means ± SEM (*n* = 10, *** *p* < 0.001). Magnified field-1 and magnified field-2 show the enlarged images of the areas in white dashed boxes 1 and 2 in the images of baicalein-treated and untreated cells, respectively. The white arrowheads in magnified field-1 indicate the colocalized signal. (**B**) Selected live imaging frames showing 0.5 mM baicalein-treated Huh7/RFP-LC3/Mito-GFP cells. The engulfed Mito-GFP shown within RFP-LC3 puncta is indicated by white arrowheads. All frames in the sequence showing this engulfment are presented in Appendix A.

**Figure 6 cells-11-01132-f006:**
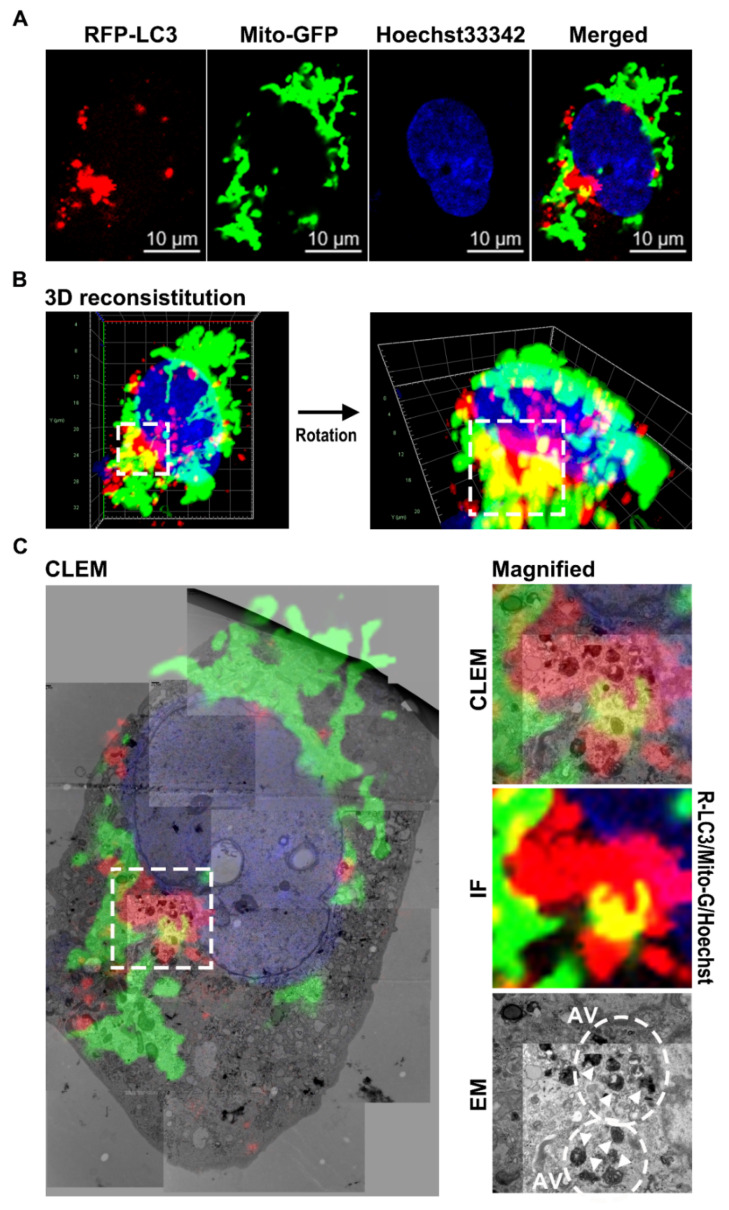
Correlative light and electron microscopy (CLEM) ultrastructural analysis of mitochondrial engulfment during autophagy in baicalein-treated cells: (**A**) Huh7/RFP-LC3/Mito-GFP cells were treated with 0.5 mM baicalein and observed with confocal microscopy. (**B**) Images from different Z-stack positions were assembled, and a deconvolution 3-D structure was created as described in “Materials and Methods” (left panel). The rotated image of this 3-D structure is shown in the right panel. The white dashed box shows the sequestration of Mito-GFP within an area with RFP-LC3-labeled puncta. (**C**) The aligned images obtained by immunofluorescence (IF) and electron microscopy (EM) in CLEM are shown in the left panel. The magnified images in the right panel show the enlargement of the area in the white dashed box of the left panel. The white dashed circles and white arrowheads indicate autophagic vacuoles (AV) and engulfed mitochondria, respectively.

**Figure 7 cells-11-01132-f007:**
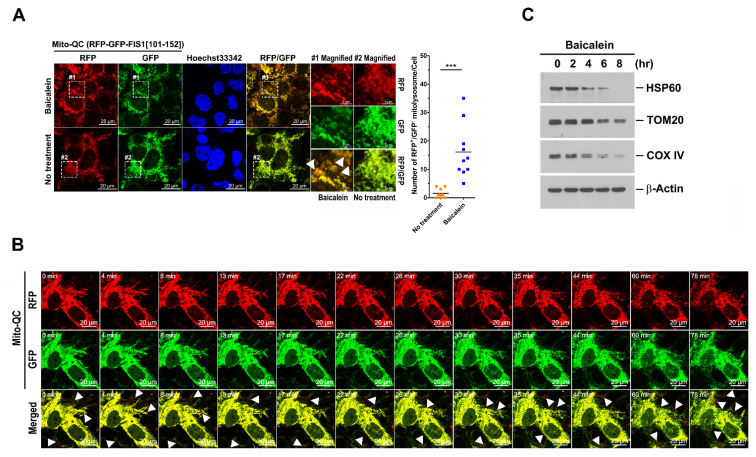
Mitophagic degradation induced by baicalein: (**A**) Huh7 cells stably expressing Mito-QC (Huh7/Mito-QC) were established by lentiviral gene delivery as described in “Materials and Methods”. The cells were treated with or without 0.5 mM baicalein for four hours, fixed, stained with Hoechst 33342, and analyzed by confocal microscopy. The white arrowheads in magnified field-1 indicate RFP^+^/GFP^−^ mitolysosomes. The number of RFP^+^/GFP^−^ mitolysosomes shown was quantified. The data represent the mean ± SEM (*n* = 10, *** *p* < 0.001). Magnified field-1 and magnified field-2 show the enlarged images of the areas in white dashed boxes 1 and 2 in the images of baicalein-treated and untreated cells, respectively. (**B**) Selected live imaging frames showing 0.5 mM baicalein-treated Huh7/Mito-QC cells (Appendix A). The white arrowheads indicate the formation of RFP^+^/GFP^−^ mitolysosomes. (**C**) Huh7 cells were treated with 0.5 mM baicalein and harvested at different times for use in the Western blot analysis.

**Figure 8 cells-11-01132-f008:**
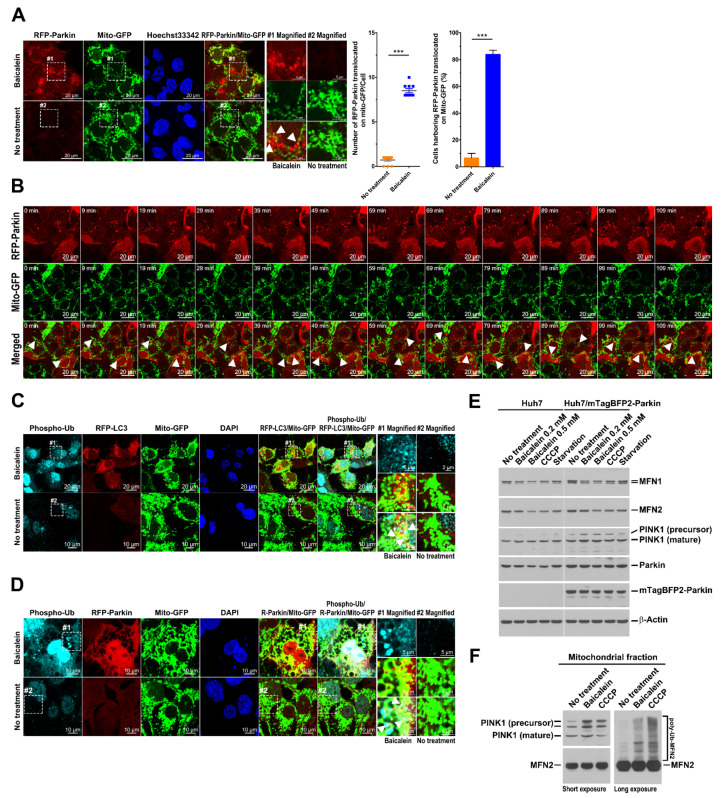
Mitochondrial translocation of Parkin and phosphorylation of ubiquitin in baicalein-treated cells: (**A**) Huh7 cells coexpressing RFP-Parkin and Mito-GFP (Huh7/RFP-Parkin/Mito-GFP) were established by lentiviral gene delivery as described in “Materials and Methods”. The cells were treated with or without 0.5 mM baicalein for four hours, fixed, stained with Hoechst 33342, and analyzed by microscopy. The number of RFP-Parkin translocated into the Mito-GFP was quantified. The data represent the means ± SEM (*n* = 10, *** *p* < 0.001). The percentage of cells containing RFP-Parkin translocated into the Mito-GFP was quantified, and the data represent the means ± SEM of three independent experiments (*** *p* < 0.001). Magnified field-1 and magnified field-2 show enlarged images of the areas in white dashed boxes 1 and 2 in the images of baicalein-treated and untreated cells, respectively. The white arrowheads in magnified field-1 indicate the overlapping signal. (**B**) Selected frames obtained by live imaging of 0.5 mM baicalein-treated Huh7/RFP-Parkin/Mito-GFP cells (Appendix A) are shown. The translocation of RFP-Parkin to Mito-GFP-labeled areas is indicated by white arrowheads. (**C**) Huh7/RFP-LC3/Mito-GFP cells were treated with or without 0.5 mM baicalein for four hours. Then, the cells were fixed, immunostained for phospho-ubiquitin (Ser65) (cyan) and nuclei by DAPI (blue), and analyzed by confocal microscopy. (**D**) Treatment with baicalein, immunostaining for phospho-ubiquitin and nuclei, and confocal microscopy analysis of Huh7/RFP-Parkin/Mito-GFP cells were performed as described in (**C**). (**E**) Huh7/mTagBFP2-Parkin cells were established by lentivirus gene delivery as described in “Materials and Methods”. Huh7 and Huh7/mTagBFP2-Parkin cells were treated with 0.2 mM baicalein and 0.5 mM baicalein for four hours, and with 10 μM CCCP and starvation medium for two hours. Cells were harvested for analysis of protein expression using Western blotting. (**F**) Huh7 cells were treated with 0.5 mM baicalein and 10 μM CCCP as shown in (**E**) and then used in the isolation of the mitochondrial fraction. Then, purified mitochondrial fractions were solubilized and analyzed for protein expression using Western blotting.

**Figure 9 cells-11-01132-f009:**
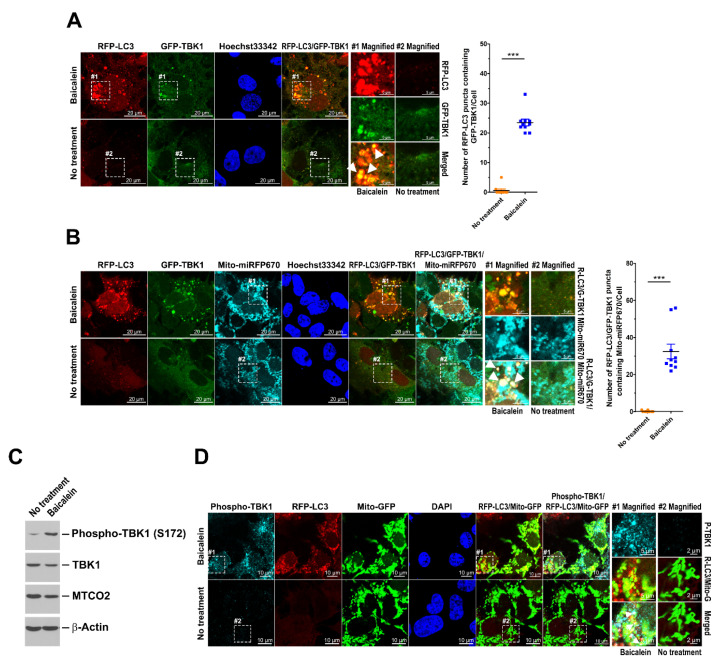
Activation of TBK1 in baicalein-induced mitophagy: (**A**) Huh7 cells stably coexpressing RFP-LC3 and GFP-TBK1 (Huh7/RFP-LC3/GFP-TBK1) were established by lentiviral gene delivery as described in “Materials and Methods”. Huh7/RFP-LC3/GFP-TBK1 cells were treated with or without 0.5 mM baicalein for four hours, fixed and stained with Hoechst 33342, and analyzed by confocal microscopy. The number of RFP-LC3 puncta colocalized with GFP-TBK1 was quantified. The data represent the means ± SEM (*n* = 10, *** *p* < 0.001). Magnified field-1 and magnified field-2, respectively, show the enlarged images of the areas in white dashed boxes 1 and 2 in the images of baicalein-treated and untreated cells. The white arrowheads in the magnified field-1 indicate the overlapping signal. (**B**) Lentiviruses harboring RFP-LC3, GFP-TBK1, and Mito-miRFP670 were transduced into Huh7 cells, generating Huh7/RFP-LC3/GFP-TBK1/Mito-miRFP670 cells. Huh7/RFP-LC3/GFP-TBK1/Mito-miRFP670 cells were cultured in the presence or absence of 0.5 mM baicalein. Four hours later, the cells were fixed, stained with Hoechst 33342, and analyzed by confocal microscopy. (**C**) Huh7 cells were treated with or without 0.5 mM baicalein for eight hours and harvested for protein expression analysis by Western blotting. (**D**) Baicalein-treated Huh7/RFP-LC3/Mito-GFP cells were fixed, immunostained for phospho-TBK1 (Ser172) and nuclei by DAPI, and analyzed by confocal microscopy.

**Figure 10 cells-11-01132-f010:**
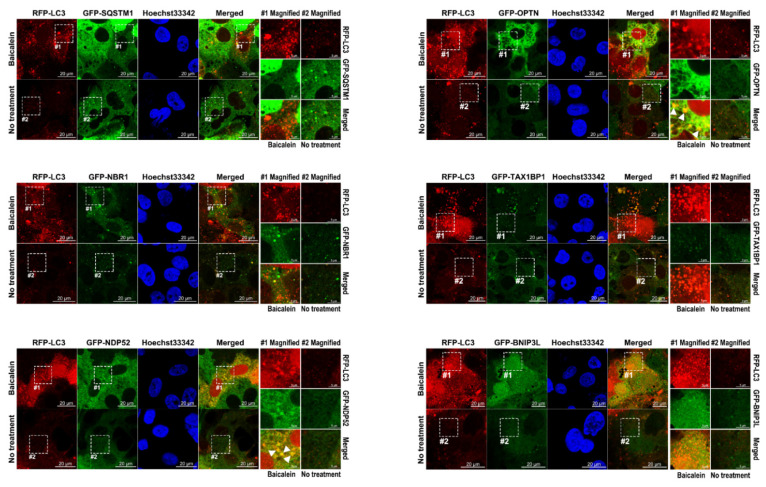
Analysis of the colocalization between cargo receptors and autophagic vacuoles in baicalein-treated cells. Huh7/RFP-LC3 cells were transduced with lentiviruses harboring GFP-SQSTM1, GFP-NBR1, GFP-NDP52, GFP-OPTN, GFP-TAX1BP1, and GFP-BNIP3L to generate Huh7/RFP-LC3/GFP-SQSTM1, Huh7/RFP-LC3/GFP-NBR1, Huh7/RFP-LC3/GFP-NDP52, Huh7/RFP-LC3/GFP-OPTN, Huh7/RFP-LC3/GFP-TAX1BP1, and Huh7/RFP-LC3/GFP-BNIP3L cell lines, respectively. The cells were treated with or without 0.5 mM baicalein for four hours. Then, the cells were fixed, stained with Hoechst 33342, and analyzed by confocal microscopy. Magnified field-1 and magnified field-2 in each panel show the enlarged images of the areas in white dashed boxes 1 and 2 in baicalein-treated and untreated cells, respectively. The white arrowheads in magnified field-1 indicate the colocalized signal.

**Figure 11 cells-11-01132-f011:**
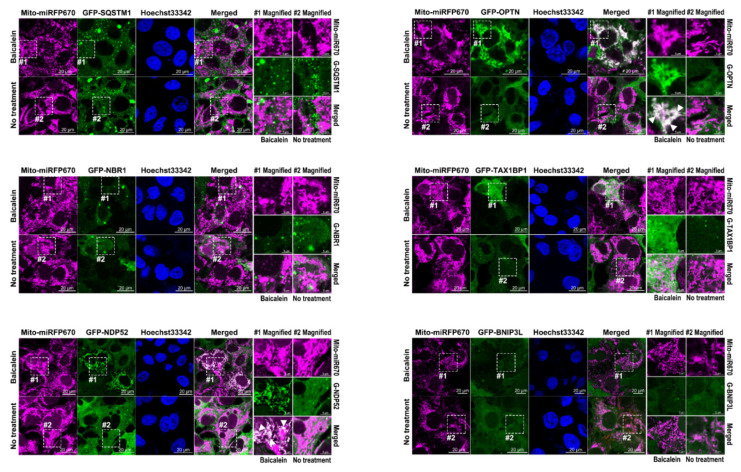
Assessment of the translocation of cargo receptors to mitochondria in baicalein-treated cells. Huh7/Mito-miRFP670/GFP-SQSTM1, Huh7/Mito-miRFP670/GFP-NBR1, Huh7/Mito-miRFP670/GFP-NDP52, Huh7/Mito-miRFP670/GFP-OPTN, Huh7/Mito-miRFP670/GFP-TAX1BP1, and Huh7/Mito-miRFP670/GFP-BNIP3L cells established by lentiviral gene delivery were cultured in the presence or absence of 0.5 mM baicalein for four hours. Then, the cells were fixed, stained with Hoechst 33342, and analyzed by confocal microscopy. Magnified field-1 and magnified field-2, respectively, show enlarged images of the areas in white dashed boxes 1 and 2 in the images of baicalein-treated and untreated cells in each panel. The white arrowheads in magnified field-1 indicate the overlapping signal.

**Figure 12 cells-11-01132-f012:**
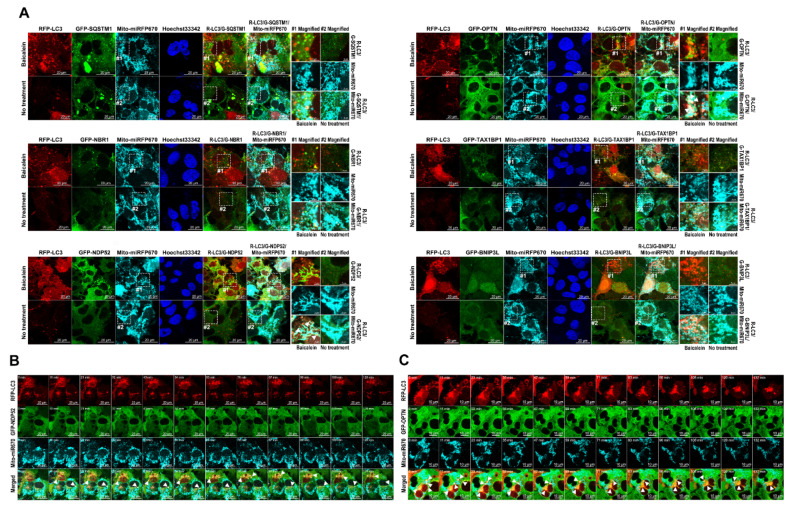
Recruitment of cargo receptors in baicalein-induced mitophagy: (**A**) Huh7/RFP-LC3/Mito-miRFP670 cells expressing individual GFP-cargo receptors as indicated were established. The cells were treated with or without 0.5 mM baicalein. Four hours later, the cells were fixed, stained with Hoechst 33342, and analyzed by confocal microscopy. Magnified field-1 and magnified field-2 in each panel show enlarged images of the areas in white dashed boxes 1 and 2 in the images of baicalein-treated and untreated cells, respectively. The white arrowheads in magnified field-1 indicate the colocalized signal. (**B**) Selected frames showing live imaging of 0.5 mM baicalein-treated Huh7/RFP-LC3/Mito-miRFP670/GFP-NDP52 cells. All frames in the live imaging sequence are shown in Appendix A. In the representative image, the arrowheads indicate the recruitment of GFP-NDP52 in the elimination of Mito-miRFP670 by RFP-LC3 puncta-labeled autophagic vacuoles. (**C**) Selected frames obtained from the live imaging analysis of 0.5 mM baicalein-treated Huh7/RFP-LC3/Mito-miRFP670/GFP-OPTN cells (Appendix A) are shown. In the representative image, the arrowheads indicate the coordination of GFP-OPTN with RFP-LC3 puncta-labeled autophagic vacuoles in the turnover of Mito-miRFP670.

**Figure 13 cells-11-01132-f013:**
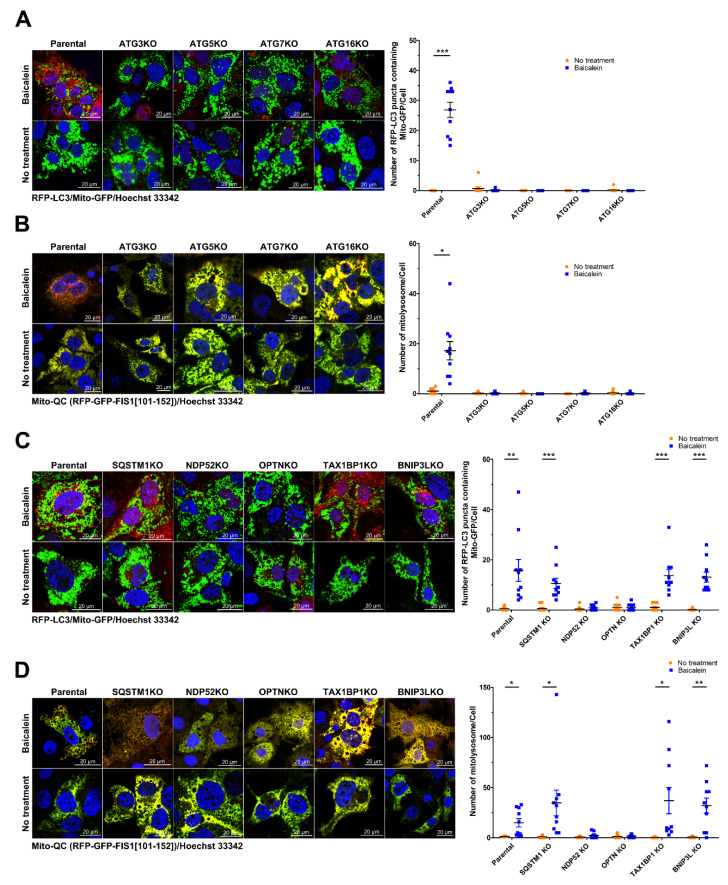
Effects of knocking out the gene expression of ATGs and cargo receptors on baicalein-induced mitophagic degradation: (**A**,**B**) Huh7 parental and individual ATG-KO cells coexpressing RFP-LC3 and Mito-GFP in (**A**) and expressing Mito-QC in (**B**) were treated with or without 0.5 mM baicalein. Four hours later, the cells were fixed, stained with Hoechst 33342, and analyzed by confocal microscopy. The number of RFP-LC3 puncta with Mito-GFP (**A**) and the number of mitolysosomes (**B**) were quantified. The data represent the means ± SEM (*n* = 10, * *p* < 0.05, ** *p* < 0.01, *** *p* < 0.001). (**C**,**D**) Huh7 parental and individual cargo receptor-KO cells stably expressing RFP-LC3 and Mito-GFP in (**C**) and expressing Mito-QC in (**D**) were cultured in the presence or absence of 0.5 mM baicalein for four hours. Then, the cells were fixed, stained with Hoechst 33342, and analyzed by confocal microscopy. The number of RFP-LC3 puncta with Mito-GFP (**C**) and the number of mitolysosomes (**D**) were quantified. The data represent the means ± SEM (*n* = 10, * *p* < 0.05, ** *p* < 0.01, *** *p* < 0.001).

## Data Availability

The data presented in this study are available on reasonable request from the corresponding author.

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
