# Peer review of "Baicalein Activates Parkin-Dependent Mitophagy through NDP52 and OPTN"

_cells, 2022, doi:10.3390/cells11071132_

Round 1
Reviewer 1 Report
This is a tremendous paper. Authors have gone to incredibly extensive lengths to demonstrate that baicalein is not just a potent autophagy inductor, but also a mitophagic one. Data is extremely well described and experimental design is flawless. I really had a very hard time finding any eventual minute flaws. Of particular note, CLEM data is fascinating.
The only major issue I have with this work is a conceptual one. While it is unbelievably detailed, it does no present a major contribution. It sort of just validates the already known mechanisms, with baicalein thrown in the mix.
However, the authors should not feel that I believe that their work is without merit, far from it. So much so that I am recommending a few minor revisions before publication.
The most important is that the introduction is far too long. You risk loosing the reader. Half of it is a revision paper, and the other half is reporting the data incoming. Both sections should be extensively trimmed down, in particular the second part of data description. In fact, these are not required to be removed, but placed elsewhere. Most of the first half could end up in the discussion, while the second one is mostly a repetition of the results section, and should be trimmed down on a per-case basis.
Other than that, so very minor grammatical corrections are required, nothing too problematic. There are also some minor inconsistencies here and there (for instance, on Fig.3C, the microscopy images have different scale bar font sizes - things like this nothing major).
Author Response
Dear reviewer:
Thank you for giving me the opportunity to resubmit my manuscript “Baicalein activates Parkin-dependent mitophagy through NDP52 and OPTN” to Cells (Manuscript ID: cells-1608165). I appreciate the thoughtful and constructive suggestions provided by the reviewer. The content of this manuscript has been improved based on the reviewer’s comments, and I have incorporated additional results showing that baicalein induces mitochondrial fragmentation and clustering, activates PINK1 stabilization in mitochondria, and promotes MFN1 and MFN2 degradation. Moreover, we also provided evidence that baicalein leads to an increase in the number of mitophagic cells containing acidic MT-Keima in a similar fashion as CCCP, a well-defined inducer of PINK1/Parkin-dependent mitophagy. In addition, we have revised and consolidated the content of the Introduction and Discussion sections according to the reviewer’s comments. The changes are shown in the revised manuscript, and point-by-point responses to each comment are listed below.
Point 1: This is a tremendous paper. Authors have gone to incredibly extensive lengths to demonstrate that baicalein is not just a potent autophagy inductor, but also a mitophagic one. Data is extremely well described and experimental design is flawless. I really had a very hard time finding any eventual minute flaws. Of particular note, CLEM data is fascinating.
Response 1: We appreciate the reviewer’s recognition of our study and commendation of our results, particularly for the CLEM study. Thank you again for your comments.
Point 2: The only major issue I have with this work is a conceptual one. While it is unbelievably detailed, it does no present a major contribution. It sort of just validates the already known mechanisms, with baicalein thrown in the mix. However, the authors should not feel that I believe that their work is without merit, far from it. So much so that I am recommending a few minor revisions before publication.
Response 2: We thank the reviewer for the thoughtful comments on our manuscript and recognition of the merit of our manuscript. We appreciate the reviewer’s recommendation on consideration of this manuscript for publication in Cells. Although PINK1/Parkin-dependent mitophagy has been extensively studied in HeLa cells ectopically expressing Parkin, whether and how mitophagy is solely regulated by the PINK1/Parkin-dependent pathway in liver cells are still unclear. Additionally, the mitophagy inducer potentially used in liver cells is still limited. In this study, we aimed to demonstrate that baicalein induces hepatic mitophagy in hepatic cells through mitochondrial translocation of Parkin and two cargo receptors, NDP52 and OPTN. In addition, through time-lapse live cell imaging and CLEM ultrastructural analyses, we provide compelling evidence showing the dynamics of baicalein-activated hepatic mitophagy. In the revised manuscript, we further provided additional studies showing that baicalein treatment led to mitochondrial fragmentation and clustering (Supplemental Figure 6), and baicalein triggered MFN1 and MFN2 degradation (Figure 8E in the revised manuscript). In addition, we demonstrated that baicalein induced PINK1 stabilization in mitochondria (Figure 8F in the revised manuscript). Moreover, we confirmed baicalein-activated mitophagy through confocal microscopy and FACS analyses of the MT-Keima reporter (Supplemental Figures 10 and 15). Together, we conclude that baicalein is a novel inducer of hepatic mitophagy activation. We hope that the reviewer kindly agrees with us to publish our work in Cells. Thank you again for the constructive comments.
Point 3: The most important is that the introduction is far too long. You risk loosing the reader. Half of it is a revision paper, and the other half is reporting the data incoming. Both sections should be extensively trimmed down, in particular the second part of data description. In fact, these are not required to be removed, but placed elsewhere. Most of the first half could end up in the discussion, while the second one is mostly a repetition of the results section, and should be trimmed down on a per-case basis.
Response 3: Thank you very much for the thoughtful suggestions on the Introduction section; and we have revised the content of the Introduction section. For the first half of the introduction, we have shortened the background of autophagy and mitophagy regulation (Please see line 30 on page 1 to line 104 on page 3 in the revised manuscript). In addition, we have reduced the content describing our results to a paragraph in the revised manuscript (Please see lines 122 to 140 on page 3 in the revised manuscript). Thank you again for the constructive comments.
Point 4: Other than that, so very minor grammatical corrections are required, nothing too problematic. There are also some minor inconsistencies here and there (for instance, on Fig.3C, the microscopy images have different scale bar font sizes - things like this nothing major).
Response 4: We are very grateful for the reviewer’s thoughtful comment on this manuscript. We have re-edited the content of the revised manuscript and fixed the grammatical errors. The editing certificate of the revised manuscript by American Journal Experts (AJE) (verification code 2FF8-E35E-73C6-F92B-BB6P) can be found in AJE website and also provided in non-published materials. We apologize for the different scale bar and font sizes in the presentation of Figure 3C. We have revised the representative images of Figure 3C with the same scale bar and font sizes (please see Supplemental Figure 4 in the revised manuscript). Thank you again for the constructive suggestions.
We hope that this version of our manuscript and our responses address all your concerns and that this revised manuscript meets the criteria for publication in Cells. Thank you for your kind consideration.
Sincerely,
Po-Yuan Ke, Ph.D.
Assistant Professor
Department of Biochemistry & Molecular Biology and Graduate Institute of Biomedical Sciences, College of Medicine, Chang Gung University, Taoyuan 33302, Taiwan, Republic of China
Liver Research Center, Chang Gung Memorial Hospital, Linkou, Taoyuan 33305, Taiwan, Republic of China
Tel: 886-3-2118800-5115
E-mail: pyke0324@mail.cgu.edu.tw
Reviewer 2 Report
While baicalein has been shown to induce various responses such as apoptosis, cell cycle arrest etc and display other pleiotropic effects including antioxidant, anti-inflammatory, its effect on autophagy/mitophagy is not well known and this manuscript aims to dig into this important process.
While I was initially overwhelmed by so many high quality figures (15!!!) and each figure contains many panels, I soon realized that many figures are just repetition of similar results with slightly different markers to show the same thing over and over. They need to be completely reorganized and cut down.
Major point:
- If baiclein induces Parkin-dependent mitophagy, then the obvious and important question will be if it is also PINK1 dependent. Does baiclein trigger mitochondrial membrane potential loss to stabilize PINK1? If it is, it will act like CCCP. Even with CCCP treatment and Parkin overexpression in HeLa cells, significant degradation of mitochondrial matrix proteins such as COXII is only seen after 18-24 hr. However, Fig. 7G shows that with balicalein treatment for 8 hrs, HSP60 and COXIV proteins are almost completely degraded. This is unheard of. The authors should expand on this discovery to validate. For instance, compare baicalein treatment with CCCP, with or without Parkin overexpression. Check PINK1 protein expression and Mfn1/2 degradation. Parkin mediated mitophagy triggers rapid Mfn1/2 degradation within an hour via UPS pathway. CCCP treatment often induces mitochondrial fragmentation and clustering when Parkin is overexpressed. Does baicalein induce similar changes?
2. Many figures are repeated with GFP-LC3, RFP-LC3, GFP-RFP-LC3, LAMP1 and LAMP2 imaging and live imaging. It's just too much. Just show GFP-RFP-LC3 data is enough. One common theme is that for all images in the control panel, the GFP, or RFP signal is significantly dimmer (barely seen) than the treated cells. This does not make sense. You expect to see similar GFP or RFP signal in both samples, the only difference is whether it is diffuse in cytosol or forms puncta. Do the authors intentionally lower the signal intensity in the control cells? Many live imaging figures are not needed as they don't provide any additional information.
3. They used three figures to show baicalein induces autophagy, then with the rest of figures to show it's mitophagy. It's not clear if baicalein induces mainly autophagy with random non-selective mitophagy or exclusively the selective mitophagy. They have the mt-Keima marker for sensitive and quantitative mitophagy measurement but didn't expand on this. They should do FACS analysis with mt-Keima to show mitophagy with starvation, baicalein and CCCP treatment with or without Parkin overexpression to show if baicalein specifically induces mitophagy, not general autophagy. Once they confirmed it, they can do it in their ATG3/5/7/16 KO, and OPTN/NDP52 KO etc to understand the mechanism.
Minor points:
1. Fig. 1A. Just show one cell type and put the other two cell lines into supplementary.
2. Fig. 2. Just show one panel of LAMP1 or LAMP2 staining, no need to show both. No need for 2D-F. 2G. LC3B-II level is not much different between untreated and baicalein treated cells. In many of WB figures regarding LC3 lipidation, it seems that LC3-I level does not decrease along with LC3-II level increase. This does not make sense.
3. Fig. 3A. Need to show WB with anti-ATG3/5/7/16 to show the validation of the KO cells. Based on the method description, all KO cells are pooled, not from individual single clone. This might explain why LC3 lipidation is still observed in ATG3/16 KO cells. 3C can be removed.
4. Fig. 4 can be condensed into one panel. Pick A or B and remove C-E.
5. Fig. 8A. Parkin recruitment should be counted as the percentage of cells showing Parkin translocation instead of count the number of Parkin foci.
6. All the CLEM figures can be removed.
7. Fig. 5 can be removed.
8. Fig. 7. no need to show both Mito-QC and mito-Keima.
9. Fig. 8. no need to show RFP-Parkin and YFP-Parkin. Pick one panel.
10. Introduction should be cut down especially with the last two paragraphs showing too much details on their results.
11. Discussion should be cut down as well as many sentences have been repeated in Introduction or somewhere else.
